# Agricultural Resources and Trade Strategies: Response to Falling Land-to-Labor Ratios in Malawi

**Sarah Ephrida Tione** 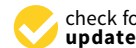

School of Economics and Business, Norwegian University of Life Sciences, P.O. Box 5003, 1432 Aas, Norway; sarahti@nmbu.no

**Abstract:** This study assesses how growing land scarcity relative to family labor is influencing farm household decisions to trade in agricultural land and labor markets to improve their livelihood. Using the farm household model, I analyze decisions to rent-in land or hire out labor among smallholders in Malawi. I use data from two rounds of a nationally representative balanced-household panel and apply a systems approach to jointly estimate land rental and labor market decisions while controlling for simultaneity and unobserved heterogeneity. The results indicate that the falling owned-land-to-labor-endowment ratio can push households to participate in either land rental or seasonal agricultural labor markets. However, the probability of hiring out labor for casual work and short-term gains decreases when potential tenant households rent-in land. Based on asset-wealth-to-labor-endowment ratios, wealthier households are more likely to rent-in land while poorer households, including most smallholder households, are more likely to hire out labor. These results suggest higher friction in the land rental market compared to the agricultural labor markets and liquidity constraints dictating what is necessary to support agricultural operations and household needs. Accordingly, agricultural policy in Malawi should aim to reduce friction in factor markets.

**Keywords:** land scarcity; factor markets; *ganyu*; conditional mixed process; Malawi

## 1. Introduction

Agricultural land in sub-Saharan Africa (SSA) has generally trended from abundance towards scarcity, mainly from population growth [1,2]. Land scarcity is particularly challenging for farm households in SSA that rely on land and labor factors of production for income and food security [3,4], with the extant literature indicating a declining owned-land-to-labor-endowment ratio among farm households, especially in areas of SSA with high population densities [5,6]. This change has resulted in farm households using more family labor relative to their farmland due to imperfections in the labor market and limited opportunities to trade their labor outside the agricultural sector [5,7]. Such labor usage would be considered less profitable if households could trade in the labor and land factor markets [7]. Therefore, understanding the choices made by farm households is important for developing policies that can improve factor markets in SSA.

The literature on household decisions——according to both the farm-household model and livelihood approach studies——demonstrate that farm households in SSA have been responding to the land-scarcity challenge by diversifying and intensifying the use of land and labor factors of production, mainly within the agricultural sector [5,8]. However, there is still limited empirical evidence regarding the extent to which changes to the owned-land-to-labor-endowment ratio influence the decision to trade in either agricultural land or labor markets as a livelihood strategy. For most countries in SSA, this empirical gap has been partly explained by missing or absent or thin land markets despite non-missing but imperfect agricultural labor markets [5].

However, land markets are developing, featuring more active and expanding land-rental markets [4,9]. Thus, this paper aims to understand how the changes in land-to-labor ratio are influencing the trade of these resources among smallholders, amidst land scarcity challenges. Specifically, I focus on households with more labor relative to owned farmland, who are capable of renting-in land or hiring out their labor for casual work in seasonal agricultural labor markets. I chose to focus on these households because of increased farm-household-level population pressure and landlessness that is leaving households with relatively more labor to owned farmland in most countries in SSA [2,3]. I analyze how the falling land-to-labor ratio affects entry into land rental and seasonal agricultural labor markets, as well as the extent of participation in either.

Furthermore, considering that land rental markets develop to enhance the efficient allocation of land and non-land factors of production [9], I assess the extent to which renting-in agricultural land influences decisions to hire out labor for seasonal agricultural work. This is because the majority of smallholder households in SSA use or trade their labor within the agricultural sector by undertaking casual work due to limited skills, capital and labor opportunities in the non-agricultural sectors [10,11]. Notably, trading labor for casual agricultural work is considered a short-term economic response to food consumption needs in SSA [10,12–14]. Casual work is also considered a coping strategy to idiosyncratic shocks that might reallocate labor away from family farms, exacerbating the poverty gap [14–17]. Accordingly, understanding how land rental markets influence labor trade decisions is key to developing policy strategies that can sustain smallholder livelihoods and improve their labor-use efficiency for medium- to long-term gains.

I use two rounds of balanced household panel data from the Malawi Living Standards Measurement Surveys (LSMS) conducted in 2013 and 2016. Malawi is appropriate for this study because it continues to experience increase in land pressure and demand for land for both food security and economic growth [6,18]. Additionally, the reviewed literature on Malawi indicates a history of farm households responding to land scarcity challenges by undertaking casual work in the localized seasonal agricultural labor markets as a strategy for coping with food consumption shocks [16,19]. Furthermore, recent evidence shows that land rental markets developing in Malawi are positively impacting resource allocation, income and welfare [20,21]. However, based on the reviewed literature, no study has yet focused on the joint decision to trade either agricultural land or labor in response to the falling land-to-labor ratio in the country. There is also limited evidence of how developing rental markets are prompting smallholder households to hire out labor for casual work.

To achieve this paper's objective, I assess entry into the land rental and seasonal agricultural labor markets using the bivariate and recursive bivariate models for joint and sequential decisions. For the extent of participation, I use a simultaneous system approach combining the Tobit model for renting-in land and a fractional probit model for hiring out labor. I apply the conditional mixed process (CMP) estimation method to consider issues of simultaneity and endogeneity, even in a recursive system [22]. Ultimately, this paper contributes new empirical evidence to the growing literature on diversification of livelihood strategies and efficient allocation of productive resources through market assisted strategies in SSA [9,23].

Following this section, I present background information on the land and seasonal agricultural labor markets in Malawi. Then, I discuss the theoretical framework and propose hypotheses in Section 2. In Section 3, I present data and estimation methods before presenting the results in Section 4. I discuss these results in Section 5 and conclude the paper in Section 6.

*Background*

Malawi is an agriculture-based country where 84 percent of the population resides in rural areas and the agricultural sector employs 64 percent of the workforce [18,24]. The country is populated at high density, reportedly at 186 persons per square kilometer and is ranked amongst the top ten populated countries in SSA [18,25]. There are 9.8 million hectares (ha) of land available in Malawi, of which 7.7 million ha are suitable for agriculture. Malawi's agricultural policy mainly focus on

the dualism of large or estate farms and smallholders. The estate sub-sector in Malawi occupies up to 1.5 million ha while smallholders occupy at least 4.5 million ha, after adjusting for wetlands, steep slopes and protected traditional land [26]. A recent study of large-scale farms in Malawi showed that almost 93 percent of estate owners hold between 10 and 30 ha, with 6 percent holding area larger than 50 ha [27]. In contrast, smallholders hold an average of one ha per household. Similar to most African countries, there are no standard definitions of large and small farm size in Malawi [28–30]. Studies have used either 10 or 20 ha as the cut-off for large farms [27,28]. In recent literature on the rise of medium-scale farms in Malawi, 5 ha has been used as the cut-off for defining small farms [28,30]. Despite the lack of clear cut-off point on farm size, LSMS and Food and Agriculture Organization (FAO) data on Malawi shows that almost 70 percent of farms are smaller than 5 ha and operated by almost two million farm households [28]. These statistics make clear that land continues to be scarce amidst population pressure [31].

In Malawi, available land is mainly governed using one of three tenure systems: customary, public, or private. Public land is the unallocated land under the control of the government and private tenure defines land under freehold and leasehold titles, while customary tenure defines community land allocated to community members or families. Customary land is held in trust to the traditional authority or local leaders to manage and redistribute according to lineage. Although both patrilineal and matrilineal lineage systems govern land inheritance, this does not directly translate to patriarchal and matriarchal systems of resource control at the household level [32–34]. The 2002 Malawi National Land Policy indicates that, under the customary tenure system, "families and individuals are allocated exclusive fee simple usufruct in perpetuity subject to effective utilization" [26]. That is, customary land remains a community property and never an individual private property. To promote tenure security, the Customary Land Act, 2016, indicates that the government will register all customary land as private customary estate at the traditional authority or community level. However, with individual efforts, households are allowed to register their land as private customary estate [35].

In Malawi, almost 67 percent of the total agricultural land is under customary tenure and mainly cultivated by smallholders in rural areas [26]. Rural households—and, to some extent, urban dwellers—depend considerably on customary land and family labor for both income and food security [6,19]. Demand for land has increased over time due to both household-level population pressure and proximity to urban areas, which have also changed how farm households value their land [36]. The literature also indicates that smallholder households in Malawi have been mainly responding to the land scarcity challenge by hiring out family labor for casual work in seasonal agricultural labor markets, commonly known as *ganyu* [16,19]. This is mostly to overcome the imbalance between land ownership and ability or desire to cultivate agricultural land, an imbalance produced by non-existent land rental markets [19]. However, considering land rental markets are now developing [21], assessing farm household trade response strategies is important for policy development in Malawi, as well as proving lessons relevant to other countries in SSA.

According to Whiteside [16], the word *ganyu* refers broadly to any "off-own-farm work" that is paid as piecework and usually agriculture-related and that individuals undertake casually, with days or weeks being paid in cash or in-kind (e.g., food). Generally, *ganyu* involves a daily wage or a short-term wage contract. Such contracts are mostly localized within neighboring communities and involve unskilled work such as plowing or weeding (using a hand hoe) or harvesting or shelling grains or legumes. *Ganyu* differs from skilled labor work, and long-term agricultural labor contracts mostly offered under estate farms, which are commonly known as "tenant labor" in Malawi [37].

Historically, smallholders engage in *ganyu* as a coping strategy during acute food shortage periods (December to February) and when there is high demand for agricultural labor, based on the unimodal rainfall pattern between November and April [16,37,38]. Some scholarly papers have argued that households that engage in *ganyu* neglect all or part of their farms, especially female-headed households, while others have suggested that households hire out male labor, leaving female members to attend to family farms, which are possible indicators of household vulnerability to shocks and a poverty

trap [14,16,17,39–41]. However, this could also indicate limited opportunities to access farmland in previously non-existing land rental markets [19].

Recent literature shows that land rental markets in Malawi are reallocating land use from relatively land-rich and labor-poor households to relatively land-poor and labor-rich households. Evidence also indicates that farm households renting-in land are more productive and wealthier in non-land factors (capital and labor) than landlords [20,21,34]. This could be an indicator of poor farm households renting out the land due to stress, unable or unwilling to sell their land due to family ties, despite their labor endowment conditions [21]. This evidence highlights the important role of capital and labor factors of production in household decisions to trade agricultural land or labor. Accordingly, this study analyses farm household decisions to participate in land rental and seasonal agricultural labor markets, considered the two primary trade strategies in response to the falling land-to-labor ratio and the changes in asset wealth (capital) relative to labor endowment. Additionally, this paper investigates the extent to which land rental markets influence smallholders in Malawi to allocate family labor to *ganyu*.

## 2. Theoretical Framework and Hypotheses

A farm household whose objective is to maximize income utility from land and labor endowments can use all or part of its endowment on their own farm or trade these resources across farms to achieve desired resource distribution [9]. Farm households with the potential to trade can either rent-in or rent out land or hire in or hire out their labor in the short to medium term. For such a household, the income utility function can be given as $\text{Max} U = U[Y]$, where utility is a twice differentiable quasi-concave function [42]. The $[Y]$ signifies the equivalent income from both crop production output and trading of resources. Thus, a farm household endowed with land $\left(\overline{A}\right)$ and labor $\left(\overline{L}\right)$ can have their intermediate own-farm resource use functions represented as $A = \overline{A} + A^i - A^o$ and $L = \overline{L} + L^i - L^o$, where $A$ and $L$ are the levels of land and labor use on cultivated farm, $A^i$ and $A^o$ are the amount of land rented-in or out, and $L^i$ and $L^o$ are the amount of labor hired in or out. Following Singh, Squire and Strauss [42], labor endowment $(\overline{L})$ is the sum of labor time used for both work $(L_u)$ and for leisure $(L_e)$, given as $\left[\overline{L} = L_u + L_e\right]$. Thus, estimating total labor at the household level is based on the total household adult-equivalent labor that farm households can use or trade while implicitly capturing leisure time.

Using the farm household decision model, Equation (1) specifies the household income utility function.

$$\underset{A,\ A^i,\ A^o, L, L^i, L^o}{\text{Max}} U[Y] = U\left[P_q q(A, L,\ k) - \eta\left(A^i\right) + \theta(A^o) - \tau\left(L^i\right) + \varphi(L^o) - p_m M\right]$$
$$\text{and } L^i \geq 0,\ L^o \geq 0,\ A^i \geq 0,\ A^o \geq 0 \tag{1}$$

From Equation (1), the household choice variables reflect either or both land and/or labor use and trade. In the equation, $(P_q)$ indicates output prices and $q(A, L,\ k)$ signifies the production function subject to own-farm use of land *(A)*, labor *(L)* and capital *(K)* factors of production. Considering the market imperfections that characterize markets in SSA, farm households mostly face non-linear transaction costs even with linear unit prices of goods, due to either information asymmetry or transportation costs [9,43]. Thus, overall non-linear prices in the equation are provided by parameters $(\eta)$ and $(\theta)$ for renting-in and renting out the land and parameters $(\tau)$ and $(\varphi)$ for hiring-in or out labor. This assumes that market participants on the demand and supply sides encounter different transaction costs even though they may face a similar land rental unit price [44]. Finally, $(p_m M)$ describes the total cost of other marketed inputs used on the farm. For simplicity, given the long gestation period of agriculture products and risk factors, the model assumes away liquidity constraints related to credit and the output market prices associated with household preferences [44–46].

Using the duality theory, this paper's theoretical framework focuses on the income function, which is twice differentiable and quasi-convex. If prices of both output $\left(P_q\right)$ and other marketed inputs

$(p_m)$ are normalized to one, the first order conditions (FOCs) from the income function are as specified in Equations (2) to (6).

Net buyer of labor (hiring-in)

$$\frac{\partial Y}{\partial L^i} = \frac{\partial q}{\partial L^i} - \frac{\partial \tau}{\partial L^i} \leq 0 \perp L^i \geq 0 \quad \text{i.e.,} \quad \frac{\partial q}{\partial L^i} \leq \frac{\partial \tau}{\partial L^i} \text{ if } L^i \geq 0 \tag{2}$$

Net seller of labor (hiring out)

$$\frac{\partial Y}{\partial L^o} = -\frac{\partial q}{\partial L^o} - \frac{\partial \tau}{\partial L^o} \leq 0 \perp L^o \geq 0 \quad \text{i.e.,} \quad \frac{\partial q}{\partial L^o} \geq \frac{\partial \tau}{\partial L^o} \text{ if } L^o \geq 0 \tag{3}$$

Net buyer of land (renting-in)

$$\frac{\partial Y}{\partial A^i} = \frac{\partial q}{\partial A^i} - \frac{\partial \eta}{\partial A^i} \leq 0 \perp A^i \geq 0 \quad \text{i.e.,} \quad \frac{\partial q}{\partial A^i} \leq \frac{\partial \eta}{\partial A^i} \text{ if } A^i \geq 0 \tag{4}$$

Net seller of land (renting out)

$$\frac{\partial Y}{\partial A^i} = -\frac{\partial q}{\partial A^o} - \frac{\partial \eta}{\partial A^o} \leq 0 \perp A^o \geq 0 \quad \text{i.e.,} \quad \frac{\partial q}{\partial A^o} \geq \frac{\partial \eta}{\partial A^o} \text{ if } A^o \geq 0 \tag{5}$$

Non-participant (shadow value)

$$\frac{\partial \tau}{\partial L^o} < \frac{\partial q}{\partial L} > \frac{\partial \tau}{\partial L^i} \quad \text{and} \quad \frac{\partial \eta}{\partial A^o} < \frac{\partial q}{\partial A} > \frac{\partial \eta}{\partial A^i} \tag{6}$$

The FOCs show that a household will only participate in the markets if the marginal revenue of trading factors of production is greater than the marginal cost of using the resources on their farm. Based on the FOCs, farm household trade strategies in response to the land scarcity challenge can be to trade in the land or labor market only, participate in both markets or not participate in any market, depending on the owned-land-to-labor-endowment ratio and their ability to participate in the market.

Based on the FOCs, the second-order conditions are $\frac{\partial^2 q}{\partial L^{i2}} \leq \frac{\partial^2 \tau}{\partial L^{i2}}; \frac{\partial^2 q}{\partial L^{o2}} \geq \frac{\partial^2 \eta}{\partial L^{o2}}; \frac{\partial^2 q}{\partial A^{i2}} \leq \frac{\partial^2 \tau}{\partial A^{i2}}$ and $\frac{\partial^2 q}{\partial A^{o2}} \geq \frac{\partial^2 \eta}{\partial A^{i2}}$, which I simplify as $q_{L^i L^i} \leq \tau_{L^i L^i}; q_{L^o L^o} \geq \tau_{L^o L^o}; q_{A^i A^i} \leq \eta_{A^i A^i}$ and $q_{A^o A^o} \geq \eta_{A^o A^o}$. The cross derivatives include $-q_{L^i L^o}; -q_{L^o A^i}; -q_{A^i A^o}; q_{L^i A^i}; q_{L^o A^o}$ and $-q_{L^i A^o}$. Considering the model assumes non-linear prices, the Hessian matrix will, therefore, not be positive semi-definite, as (7) demonstrates. This implies that the ability to participate in the markets is further a function of non-linear transaction costs that can raise the household shadow value on factors of production, rationing potential market participants into the non-participating group. Equation (8) presents the comparative static matrix with respect to land $(\overline{A})$ and labor endowment $(\overline{L})$. Using Equations (9) and (10) for only renting-in land and hiring out labor, I note that the change in the amount of land rented-in reduces with land endowment while hiring out labor increases with labor endowment.

$$\begin{bmatrix} q_{L^o L^o} - \tau_{L^o L^o} & -q_{L^o A^i} \\ -q_{A^i L^o} & q_{A^i A^i} - \eta_{A^i A^i} \end{bmatrix} \begin{bmatrix} dL^o \\ dA^i \end{bmatrix} \tag{7}$$

$$\begin{bmatrix} \frac{\partial^2 Y}{\partial l^o \partial l} & \frac{\partial^2 Y}{\partial l^o \partial A} \\ \frac{\partial^2 Y}{\partial A^i \partial l} & \frac{\partial^2 Y}{\partial A^i \partial A} \end{bmatrix} \begin{bmatrix} d\overline{L} \\ d\overline{A} \end{bmatrix} = \begin{bmatrix} q_{L^o \overline{L}} & q_{L^o \overline{A}} \\ -q_{A^i \overline{L}} & -q_{A^i \overline{A}} \end{bmatrix} \begin{bmatrix} d\overline{L} \\ d\overline{A} \end{bmatrix} \tag{8}$$

$$\frac{\partial A^i}{\partial \overline{A}} = \frac{-q_{A^i \overline{A}}(q_{L^o L^o} - \tau_{L^o L^o}) + \left( q_{A^i L^o} * q_{L^o \overline{A}} \right)}{(q_{L^o L^o} - \tau_{L^o L^o})(q_{A^i A^i} - \eta_{A^i A^i}) - q_{A^i L^o 2}} < 0 \text{ for renting} - \text{in land} \tag{9}$$

$$\frac{\partial l^o}{\partial \overline{L}} = \frac{q_{L^o\overline{L}}(q_{A^iA^i} - \eta_{A^iA^i}) - (q_{A^i\overline{L}} * q_{L^oA^i})}{(q_{L^oL^o} - \tau_{L^oL^o})(q_{A^iA^i} - \eta_{A^iA^i}) - q_{L^oA^{i2}}} > 0 \text{ for hiring out labor} \qquad (10)$$

From the farm household model, one can assess the land-to-labor ratio by holding constant one resource variable at a time and estimating the direction of change with non-linear transaction costs [47]. Additionally, given farm households routinely make farm decisions, the static model described can be extended to dynamic decisions over time [48]. Table 1 summarizes the optimal trade response strategies for farm households, while detailed optimal trade options, which reflect the FOCs, are presented in each cell in Appendix A, Table A1. Accordingly, assessing each solution individually or in conjunction with other solutions, even for non-participating but potential market participants, is important for understanding the farm household trade responses to changing resource ratios in SSA.

**Table 1.** Summary of potential optimal household trade response strategies.

| | | Trade Response Strategies | | |
|---|---|---|---|---|
| | | Labor Option (Equation (2)) | | |
| | | Hire in ($L^i > 0$) | Non-participant ($L^i = 0 = L^o$) | Hire out ($L^o > 0$) |
| **Land option (Equation (1))** | Rent in ($A^i > 0$) | Hiring in labor or renting-in land (*Labor poor and land poor*) | Not trading labor but renting-in land (*Labor sufficient and land poor*) | Hiring out labor or renting-in land (*Labor rich and land poor*) |
| | Non-participant ($A^o = 0$ $A^i = 0$) | Hiring in labor or not trading land (*Labor poor and land sufficient*) | Trading neither labor nor land (*Labor and land sufficient*) | Hiring out labor or not trading land (*Labor rich and land sufficient*) |
| | Rent out ($A^o > 0$) | Hiring in labor or renting out land (*Labor poor and land rich*) | Not trading labor but renting out the land (*Labor sufficient and land rich*) | Hiring out labor or renting out land (*Labor rich and land rich*) |

Given increasing land pressure and assuming more family labor relative to land for the majority of farm households in Malawi, I hypothesize four statements.

**Hypothesis H1.** *Falling owned-land-to-labor-endowment ratios increase entry to and extent of (amount of land rented in) farm household participation in land rental markets.*

With land rental markets developing as efficiency-enhancing mechanisms for the non-land factor markets, this hypothesis focuses on whether the changing ratio is a push-factor among smallholders in the land rental market. A push-factor is a negative factor that may force farm households to seek additional livelihood activities within or outside the farm [8].

**Hypothesis H2.** *Entry into the land rental market is negatively associated with trading labor for casual work in seasonal agricultural labor markets.*

According to the theoretical framework, renting-in land increases operational farmland. By renting-in agricultural land, farm households can allocate more family labor to their farm instead of hiring it out for short-term wages, assuming imperfect and seasonal agricultural labor markets. Additionally, assuming that farm households make land rental decisions at the start of the season while making the labor decisions throughout the season, I use a recursive model to assess how entry into land rental markets can influence entry into labor markets.

**Hypothesis H3.** *Increases to the household-asset-wealth-to-labor-endowment ratio increases entry to and extent of participation in the land rental markets for tenant households.*

**Hypothesis H4.** *Increases to the household-asset-wealth-to-labor-endowment ratio reduces entry to and extent of hiring out labor for casual agricultural work.*

Using the asset-wealth variable should better enable assessment of the household's ability to finance agricultural activities than using available income at the farm household level. Using available

household income as a variable is complicated by the liquidity constraints that most farm households confront due to the unimodal rainfall pattern and high output price fluctuations [49]. Thus, higher asset-wealth-to-labor-endowment ratio should be associated with a farm household's higher capacity to rent-in land and reduced need to hire out labor for *ganyu*.

## 3. Data and Estimation Methods

The study uses data from the 2013 and 2016 rounds of the LSMS. From 2013 and using 5 ha cut-off for smallholders, I constructed a balanced panel of 1895 households from an original sample of 1990 households, signifying a 4 percent attrition rate [28]. Using the inverse mills ratio estimated from a probit model, I did not observe attrition bias; hence, my results exclude the inverse mills ratio. Given land and labor decisions in Malawi mainly follow rainfed production based on a unimodal rainfall pattern, farm households are also vulnerable to production shocks that vary across different agro-ecological zones [50]. Accordingly, I used monthly district-level rainfall data that were shared upon request by Malawi's Department of Climate Change and Meteorological Services (www.metmalawi.gov.mw). The monthly district-level rainfall data cover 10 years (2007–2017) and cover the within-region rainfall variations across the three regions in Malawi (North, Central, and South). I consider these variations to substantially prompt households to trade their land and labor resources as coping strategies in the context of increasing climate shocks in the country [50].

To estimate results, I jointly assessed participation in land rental and seasonal agricultural labor markets, using bivariate and recursive bivariate models for entry into the markets and using Tobit and fractional probit models for the extent of participation [22,51,52]. The Tobit model analyzed the farmland area rented at the household level (measured in ha) while the fractional probit analyzed the share of adult equivalent household labor allocated for casual work in the seasonal agricultural labor markets. The additional use of the recursive bivariate model allowed insights into how entry into land rental markets—-a decision that households make early in the season—is likely to influence the labor decisions made throughout the production season. Furthermore, given the limited dependent variables, I applied the correlated random effects (CRE) model that uses the Mundlak [53] and Chamberlain [54] device, which is equivalent to using household fixed effects in a model with a continuous dependent variable. Finally, I used Roodman's [22] conditional mixed process that applies the full-information maximum likelihood estimation method to jointly estimate the equations following Zellner's seemingly unrelated regression model specification. The STATA program files used for this analysis are available online in the supplementary materials.

Based on the farm household decision model, I jointly estimated the reduced functional form of household renting-in land or hiring out labor over time. Equations (11) to (13) specify the farm household decision to participate in the land rental ($A^i_{jt}$) and seasonal agricultural labor ($L^o_{jt}$) markets for household $j$ and at time $t$ as a joint decision. Equations (11) and (12) are for bivariate entry to and extent of participation in each market while (13) describes the recursive bivariate land and labor market decisions. The parameters of interest, based on the hypotheses, are $\beta$, $\sigma$, and $\delta$ for household land-to-labor ratio, asset-wealth-to-labor ratio and renting-in land under a recursive system, respectively.

$$A^i_{jt} = \alpha + \beta\left(\frac{\overline{A}}{\overline{L}}\right)_{jt} + \sigma\left(\frac{K}{\overline{L}}\right)_{jt} + \gamma\check{X}_j + \pi\hat{X}_j + \check{Z}_j\beta + \hat{Z}_j\delta + \lambda N^{e+m}_{t-1} + \varrho H^{e+m}_{t-1} + \tau + \mu_j + \varepsilon_{jt} \quad (11)$$

$$L^o_{jt} = \alpha + \beta\left(\frac{\overline{A}}{\overline{L}}\right)_{jt} + \sigma\left(\frac{K}{\overline{L}}\right)_{jt} + \gamma\check{X}_j + \pi\hat{X}_j + \check{Z}_j\beta + \hat{Z}_j\delta + \lambda N^{e+m}_{t-1} + \varrho H^{e+m}_{t-1} + \tau + \mu_j + \varepsilon_{jt} \quad (12)$$

$$L^o_{jt} = \alpha + \delta A^i_{jt} + \beta\left(\frac{\overline{A}}{\overline{L}}\right)_{jt} + \sigma\left(\frac{K}{\overline{L}}\right)_{jt} + \gamma\check{X}_j + \pi\hat{X}_j + \check{Z}_j\beta + \hat{Z}_j\delta + \lambda N^{e+m}_{t-1} + \varrho H^{e+m}_{t-1} + \tau + \mu_j + \varepsilon_{jt} \quad (13)$$

Land endowment $(\overline{A})$ is defined as land acquired through customary inheritance systems, government distribution and purchases. Following Holden, Otsuka and Deininger [55], I categorized households who use farmland that is borrowed, encroached-upon or farmed under estate management as controlling pre-rental farmland without secure land ownership. Thus, such households are landless in an ownership sense because they only hold tenure as a land user. Nonetheless, I controlled for pre-rental land in the analysis, with pre-rental land capturing all farmland operated by a household except for rented-in land.

For the asset-wealth-to-labor ratio, I used the factor component analysis (FCA) to estimate the asset wealth ($K$) index for each household according to their endowment of durable goods and farm implements. The list of goods and farm implements included in the factor component analysis is presented in Appendix A. For the labor endowment $(\overline{L})$, I estimated the total adult equivalent labor from all available household members in a year. Using the asset-wealth-to-labor-endowment ratio, I categorized households into four quantiles (25th quartile) reflecting their endowment and capacity to rent-in land. According to this, households in the 25th quartile were considered poor compared to those above the 75th quartile, with intermediate groups in an ascending order.

According to the CRE specification, the equations controlled for "means" and "deviations from the means" of the community or household $(\check{Z}_j \; \hat{Z}_j)$ and farm $(\check{X}_j, \; \hat{X}_j)$ characteristics. Specifically, the analysis controlled for sex, age, and education of the household head (HH), household-size-to-labor ratio, livestock ownership based on total livestock unit (TLU) in the current and one-year lag periods, pre-rental farmland, and distance to nearest city zone (for proximity to urban areas with high demand for land and a higher likelihood of labor opportunities in both agricultural and non-agricultural sectors).

Furthermore, the analysis controlled for one-year lag upside and downside rainfall variations occurring early to mid-season at the district-level and within Malawi's regions. Such rainfall variation should reflect the spatial production shock effects that can facilitate the need to shift household land and labor resources through factor markets. The variables $N_{t-1}^{e+m}$ and $H_{t-1}^{e+m}$ signify one-year lag downside *(N)* and upside *(H)* deviations, while *(e)* describes early and *(m)* describes mid-season periods. Given the unimodal rainfall pattern in Malawi, the early to mid-season variables were designed to capture the period from October to February of each production season. October was included as a preparation month, with the early to mid-season variations used to reflect the production shock effects related to crop development and maturity while excluding Malawi's harvesting time, which coincides with the late-season period [56].

Finally, $\tau$ is the time dummy while $\mu_j + \varepsilon_{jt}$ is the additive error term incorporating time constant unobserved heterogeneity $(\mu_j)$ and the independent and identically distributed idiosyncratic error $(\varepsilon_{jt})$. Although observational data limit the full estimation of causal effects, I contend that the estimation methods sufficiently accounted for potential endogeneity and simultaneity concerns regarding this analysis. Thus, interpreting the results should provide critical insights into policy issues relevant to improving factor markets in Malawi.

## 4. Results

### 4.1. Descriptive Statistics

Recall that this paper considers a lower land-to-labor ratio to imply more labor relative to farmland and a higher ratio to imply less labor relative to farmland. The Lorenz curves in Figure 1a,b show the distribution of the ratios of owned and operational land compared to labor endowments. These figures broadly categorize farm households into four groups reflecting who participated in one or both markets and who did not participate in any factor market. However, to assess the statistical difference according to the hypotheses, the four categories in the Lorenz curves are re-organized into three main categories, which are presented in Table 2.

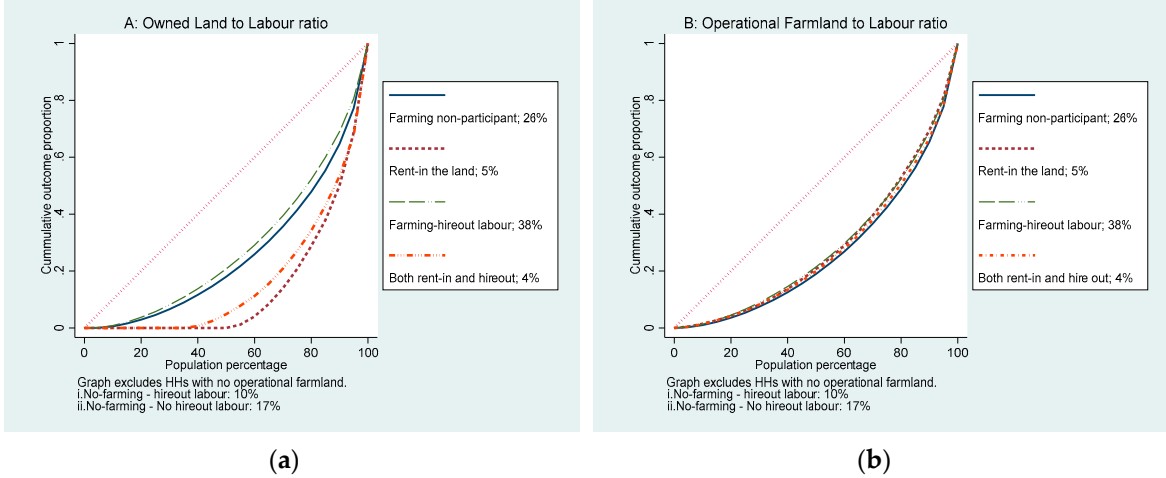

**Figure 1.** Lorenz curves: (**a**) owned-land-to-labor-endowment ratio; and (**b**) operational-farmland-to-labor-endowment ratio. Note: owned land excludes rented, borrowed, or encroached-upon land. Operational or cultivated land includes land from all reported sources at the farm household level.

The categories in Table 2 and their respective percentages in the sample are (i) renting-in or tenant households (9 percent); (ii) hiring out or casual labor households (52 percent); and (iii) non-market participant households (43 percent). Focusing on the trade response strategies in this study, the t-test in Table 2 assesses the differences between households renting-in (tenants) and hiring out (casual labor), independent of the non-market participant households. To further understand the heterogeneity in asset endowment that can reflect potential market participants within the non-market participant group, I sub-divided the households in that group into three further categories: (i) regular farmers defined as households that cultivated their land in both survey rounds; (ii) non-regular farmers defined as households that cultivated their land in only one of the survey rounds; and (ii) non-agricultural households that did not engage in cultivation during either survey round.

The Lorenz curves in Figure 1a show a relatively high owned-land-to-labor-endowment ratio for households hiring out labor and relatively low owned-land-to-labor-endowment ratio for households renting-in land. This implies that households hiring out labor for *ganyu* had less family labor relative to owned land, an indicator of hiring out of labor in distress. For tenant households, the lower ratio shown by the Lorenz curves in Figure 1a implies more family labor relative to owned land, which could be important for renting-in land [57]. The owned-land-to-labor-endowment ratios for farming non-participants were slightly lower than those for casual labor households. Although households that engaged in both markets demonstrated lower land-to-labor ratios compared to casual labor households, these ratios were slightly above those of tenant households.

The differences in the intermediate groups reflect the capacity of households to easily shift in and out of either market depending on their endowments, needs, and market-related transaction costs over time. Therefore, operational-land-to-labor-endowment ratios should reflect this shift. From Figure 1b, there are no visible differences in operational-farmland-to-labor ratios between different groups of farm households. This could signify the importance of land and labor factor markets for farm households reallocating land and labor.

Table 2 shows that there were significant differences in the ratios of households renting-in land compared to those hiring out labor for *ganyu*. However, while the table does not show significant differences for adult equivalent household labor, it does indicate significant differences in owned land between tenant and casual labor households. Thus, the differences in the owned-land-to-labor endowment ratio should reflect Malawi's land scarcity challenges. Regarding other household characteristics, Table 2 shows that tenant households owned an average of 0.37 ha but rented in an average of 0.49 ha, which increased their operational farm size to an average of 0.87 ha. The share of labor allocated to casual work averaged 28 percent of total adult equivalent household labor.

On average, casual labor households owned and operated 0.5 ha, which was significantly lower than tenant households. The percentage of households with no pre-rental farmland was higher among those renting-in land compared to casual labor households after excluding households in the non-market participant group. Furthermore, tenant households were significantly wealthier than casual labor households, as indicated by higher asset wealth index values.

Regarding the asset-wealth-to-labor-endowment ratio, tenant households were more heavily distributed in the upper quartiles while casual labor households were more heavily distributed in the lower quartiles. However, observations of the non-market participant categories indicate that the tenant households were not the wealthiest in the sample. This suggests that renting-in land might not signal the rich exploiting the poor; instead, this might be related to resource use by smallholders. Among non-market participants, although regular farmers were poor, they were generally slightly better positioned than casual labor households, indicating that poorer households were using *ganyu* as a coping strategy. The non-regular farmers and the non-agricultural households were wealthier than the tenant households, indicating less dependence on rented land and farming activities.

Furthermore, Table 2 shows a higher percentage of female-headed households among casual labor households, who are also headed by slightly older and less educated heads compared to tenant households. Casual labor households also owned relatively few livestock units compared to tenant households. There were no significant differences in household-size-to-labor ratios, demonstrating that both household categories aimed to produce for their own consumption. At the community level, distance to the nearest city zone shows that tenant households were, on average, farther from the nearest city zone than casual labor households. At the district level, the rainfall shock variables show that the one-year lag downside rainfall deviations were on average higher than the upside deviations in 2013. In contrast, in 2016, the one-year lag upside rainfall deviations were, on average, higher than downside deviations. Such rainfall variations are critical for accounting for the spatial and intertemporal production shock effects in Malawi.

*4.2. Regression Results*

Tables 3 and 4 present the key results that can be used to test the hypotheses. Specifically, Table 3 presents the bivariate and recursive bivariate average marginal effects for entry into the land rental and seasonal agricultural labor markets based on the CMP estimation method. Table 4 presents the CMP average marginal effects for the extent of participation in the land rental and casual labor markets, estimated using the Tobit and fractional probit models. In both tables, models one and two present the joint random effects models, while models three and four present the joint CRE models. The random effects models were a starting point in the analysis, which I used to check the robustness of model specifications before estimating the CRE.

Nevertheless, discussion of the results mainly focuses on the joint CRE models presented in Tables 3 and 4. The full analytical results expounding Tables 3 and 4 are presented in Appendix A, Tables A2–A5. The following control variables are collapsed in Tables 3 and 4: (i) one-year lag rainfall variations in terms of positive deviation (dm) and absolute negative deviation (dm) for early to mid-season variables; (ii) household control variables, namely sex of HH (1 = female), age of HH (years), education of HH (years), household-size-to-labor ratio, TLU-to-labor ratio, one-year lag TLU-to-labor ratio, and distance to the nearest city zone (km); and (iii) regional dummies, namely Northern, Central and Southern regions. In the following paragraphs, I present the results according to the stated hypotheses.

**Table 2.** Statistical summary.

| VARIABLES | Land and Labor Market Participant | | | Non-Market Participant (Non-Tenant and Non-Casual Labor Households) | | |
|---|---|---|---|---|---|---|
| | Tenant (1) | Casual Labor (2) | *t*-test (1 vs. 2) | Regular Farmer (Farmed in Both Survey Rounds) (3) | Non-Regular Farmer (Farmed in One Survey Round) (4) | Non-Agricultural Household (No Farming in All Rounds) (5) |
| **Land and labor participation variables** | | | | | | |
| Rent in dummy | 9.2 | | | | | |
| Rent in land (mean ha) | 0.49 (0.24) | | | | | |
| Casual labor dummy | | 51.9 | | | | |
| Share of hired out labor | | 27.5 | | | | |
| Non-participant dummy | | | | 26.3 | 3.6 | 13.3 |
| **Endowment variables** | | | | | | |
| Owned farmland (mean ha) | 0.37 (0.03) | 0.52 (0.01) | **** | 0.76 (0.02) | 0.0 | 0.0 |
| Operational farmland (mean ha) | 0.87 (0.04) | 0.58 (0.02) | **** | 0.78 (0.02) | 0.0 | 0.0 |
| No pre-rental land dummy | 41.4 | 21.5 | **** | 0.0 | 100 | 100 |
| Household labor (mean adult equiv.) | 3.38 (0.08) | 3.32 (0.03) | | 3.10 (0.05) | 3.01 (0.13) | 3.04 (0.06) |
| Owned-farmland-to-labor ratio (mean ha/adult equiv. labor unit) | 0.12 (0.01) | 0.17 (0.01) | **** | 0.28 (0.01) | 0.0 | 0.0 |
| Operational-farmland-to-labor ratio (mean ha/adult equiv. labor unit) | 0.29 (0.02) | 0.19 (0.01) | **** | 0.30 (0.01) | 0.0 | 0.0 |
| Asset wealth index (mean value) | 0.05 (0.05) | −0.30 (0.01) | **** | −0.05 (0.03) | 0.59 (0.13) | 1.06 (0.06) |
| Quartiles of asset-wealth-index-to-labor ratio | | | | | | |
| Quartile 1 | 18.1 | 29.3 | **** | 26.9 | 24.6 | 8.9 |
| Quartile 2 | 24.3 | 32.4 | *** | 22.6 | 12.3 | 7.8 |
| Quartile 3 | 30.6 | 26.1 | * | 28.3 | 17.4 | 13.7 |
| Quartile 4 | 26.9 | 12.2 | **** | 22.2 | 45.7 | 69.6 |
| **Household variables** | | | | | | |
| Sex of household head (HH) dummy (1 = Female) | 13.7 | 25.8 | **** | 27.4 | 23.9 | 18.3 |
| Age of HH (mean years) | 40.8 (0.68) | 42.8 (0.33) | ** | 47.3 (0.53) | 41.4 (1.38) | 40.3 (0.57) |
| Education of HH (mean years) | 7.2 (0.25) | 5.4 (0.09) | **** | 5.8 (0.15) | 8.4 (0.39) | 10.6 (0.23) |
| Household size to labor ratio | 1.72 (0.03) | 1.69 (0.01) | | 1.64 (0.02) | 1.53 (0.03) | 1.58 (0.03) |
| Total Livestock Units (TLU)-to-labor ratio (mean) | 0.13 (0.02) | 0.08 (0.01) | *** | 0.19 (0.03) | 0.05 (0.01) | 0.06 (0.03) |
| One-year lag TLU-to-labor ratio (mean) | 0.09 (0.14) | 0.05 (0.00) | *** | 0.08 (0.01) | 0.05 (0.02) | 0.15 (0.07) |
| Distance to the nearest city zone (mean km) | 32.7 (0.91) | 30.2 (0.43) | ** | 31.3 (0.62) | 19.8 (1.57) | 11.7 (0.64) |
| Observations (N) | 350 | 1966 | | 998 | 138 | 503 |
| **Rainfall variations (early plus mid-season)** | **2013** | **2016** | **Total** | | | |
| One-year lag positive deviation (mean dm) | 0.63 (0.01) | 1.84 (0.45) | 1.23 (0.03) | | | |
| Absolute one-year lag negative deviation (mean dm) | 0.78 (0.02) | 1.23 (0.03) | 0.94 (0.01) | | | |

Note: standard errors in parentheses. **** indicates $p < 0.001$; *** indicates $p < 0.01$; ** indicates $p < 0.05$; * indicates $p < 0.1$.

**Table 3.** Bivariate probit model with conditional mixed process (CMP) margins for land rental and casual labor (*ganyu*) market participation.

| | Bivariate Probit | | Recursive Bivariate Probit | | Bivariate Probit | | Recursive Bivariate Probit | |
|---|---|---|---|---|---|---|---|---|
| | Random Effects (CMP Margins) | | Random Effects (CMP Margins) | | Correlated Random Effects (CMP Margins) | | Correlated Random Effects (CMP Margins) | |
| | 1a | 1b | 2a | 2b | 3a | 3b | 4a | 4b |
| VARIABLES | Rent in | Hire out | Rent in | Hire out | Rent in | Hire out | Rent in | Hire out |
| **Key variables** | | | | | | | | |
| Land rented in (1 = Yes) | | | | −0.379 **** | | | | −0.375 **** |
| | | | | (0.09) | | | | (0.09) |
| Owned-farmland-to-labor ratio | −0.09 5 ** | −0.202 **** | −0.10 1 *** | −0.222 **** | −0.096 ** | −0.199 **** | −0.102 *** | −0.219 **** |
| (ha/adult equiv. labor unit) | (0.04) | (0.04) | (0.04) | (0.04) | (0.04) | (0.04) | (0.04) | (0.04) |
| Asset-wealth-index-to-labor ratio | | | | | | | | |
| Base: Quartile 4 | | | | | | | | |
| Quartile 1 | −0.037 ** | 0.236 **** | −0.045 *** | 0.209 **** | −0.038 ** | 0.213 **** | −0.046 *** | 0.188 **** |
| | (0.02) | (0.02) | (0.02) | (0.03) | (0.02) | (0.03) | (0.02) | (0.03) |
| Quartile 2 | −0.005 | 0.275 **** | −0.010 | 0.254 **** | −0.007 | 0.253 **** | −0.011 | 0.235 **** |
| | (0.02) | (0.02) | (0.02) | (0.02) | (0.02) | (0.02) | (0.02) | (0.02) |
| Quartile 3 | 0.021 | 0.196 **** | 0.019 | 0.190 **** | 0.021 | 0.178 **** | 0.019 | 0.174 **** |
| | (0.01) | (0.02) | (0.01) | (0.02) | (0.02) | (0.02) | (0.01) | (0.02) |
| No pre-rental land (1 = yes) | 0.052 **** | −0.138 **** | 0.047 *** | −0.106 **** | 0.053 **** | −0.132 **** | 0.049 **** | −0.100 **** |
| | (0.01) | (0.02) | (0.01) | (0.02) | (0.02) | (0.02) | (0.01) | (0.02) |
| **Control variables** | | | | | | | | |
| One-year lag rainfall variations | Yes | Yes | Yes | Yes | Yes | Yes | Yes | Yes |
| Observed household control variables | Yes | Yes | Yes | Yes | No | No | No | No |
| Mean of observed household variables | No | No | No | No | Yes | Yes | Yes | Yes |
| Deviations from the above mean | No | No | No | No | Yes | Yes | Yes | Yes |
| Regional dummies | Yes | Yes | Yes | Yes | Yes | Yes | Yes | Yes |
| 2016. year | −0.016 * | 0.223 **** | −0.012 | 0.202 **** | −0.017 * | 0.218 **** | −0.015 | 0.198 **** |
| | (0.01) | (0.02) | (0.01) | (0.02) | (0.01) | (0.02) | (0.01) | (0.02) |
| Constant | −1.633 **** | 0.258 | −1.541 **** | 0.281 | −1.808 **** | 0.375 * | −1.679 **** | 0.369 * |
| | (0.25) | (0.17) | (0.25) | (0.17) | (0.30) | (0.20) | (0.30) | (0.20) |
| atanhrho_12 | | −0.038 | | 0.625 *** | | −0.038 | | 0.620 *** |
| | | (0.04) | | (0.21) | | (0.04) | | (0.22) |
| N | 3790 | 3790 | 3790 | 3790 | 3790 | 3790 | 3790 | 3790 |

Note: standard errors in parentheses. **** indicates $p < 0.001$; *** indicates $p < 0.01$; ** indicates $p < 0.05$; * indicates $p < 0.1$. Full model results in Appendix A, Tables A2 and A3.

**Table 4.** Conditional mixed process (CMP) margins from a systems approach using Tobit for the land rental markets and fractional probit for the share of adult equivalent labor hired out for casual work (*ganyu*).

| | Random Effects (CMP Margins) | | Correlated Random Effects (CMP Margins) | |
|---|---|---|---|---|
| | 1 | 2 | 3 | 4 |
| VARIABLES | Tobit: Rent in | Fractional Probit: Hire out | Tobit: Rent in | Fractional Probit: Hire out |
| **Key variables** | | | | |
| Owned-farmland-to-labor ratio | −0.087 ** | −0.052 ** | −0.088 ** | −0.049 * |
| (ha/adult equiv. labor unit) | (0.04) | (0.03) | (0.04) | (0.03) |
| Asset-wealth-index-to-labor ratio | | | | |
| Base: Quartile 4 | | | | |
| Quartile 1 | −0.047 *** | 0.243 **** | −0.047 *** | 0.229 **** |
| | (0.02) | (0.02) | (0.02) | (0.02) |
| Quartile 2 | −0.014 | 0.185 **** | −0.014 | 0.173 **** |
| | (0.02) | (0.02) | (0.02) | (0.02) |
| Quartile 3 | 0.017 | 0.105 **** | 0.018 | 0.095 **** |
| | (0.01) | (0.02) | (0.02) | (0.02) |
| No pre-rental land (1 = yes) | 0.054 **** | −0.070 **** | 0.055 **** | −0.067 **** |
| | (0.01) | (0.01) | (0.01) | (0.01) |
| **Control variables** | | | | |
| One-year lag rainfall variations | Yes | Yes | Yes | Yes |
| Observed household control variables | Yes | Yes | No | No |
| Mean of observed household variables | No | No | Yes | Yes |
| Deviations from the above mean | No | No | Yes | Yes |
| Regional dummies | Yes | Yes | Yes | Yes |
| 2016. year | −0.011 | 0.108 **** | −0.013 | 0.102 **** |
| | (0.01) | (0.01) | (0.01) | (0.01) |
| Constant | −1.614 **** | −0.421 *** | −1.802 **** | −0.230 |
| | (0.25) | (0.13) | (0.31) | (0.16) |
| lnsig_1 | | −0.040 | | −0.043 |
| | | (0.06) | | (0.06) |
| atanhrho_12 | | −0.018 | | −0.018 |
| | | (0.03) | | (0.03) |
| N | 3790 | 3790 | 3790 | 3790 |

Note: standard errors in parentheses. **** indicates $p < 0.001$; *** indicates $p < 0.01$; ** indicates $p < 0.05$; * indicates $p < 0.1$. Full model results in Appendix A, Tables A4 and A5.

In H1, the study proposed that the falling owned-land-to-labor-endowment ratio increases entry to and extent of (amount of land rented in) farm household participation in land rental markets. The results indicate that an increased ratio decreases participation in land rental markets, according to both the bivariate and recursive bivariate models. This implies that a decreasing owned-land-to-labor-endowment ratio increases participation in land rental markets. According to Table 3 (models 3a and 4a), a decrease in the land per adult equivalent labor (ha/labor unit) increases entry into the land rental market by an average of 10 percentage points at the 5 percent significance level. According to Table 4 (model 3), the extent of land rental market participation also increased by an average of 0.08 hectares with a unit decrease in the land per adult equivalent labor at the 5 percent significance level.

Additionally, both the bivariate and recursive bivariate models in Table 3 demonstrate that, at 1 percent significance level, the decrease in the owned-land-to-labor-endowment ratio (ha/labor unit) was also likely to increase the hiring out of labor for *ganyu* by an average of 20 percentage points. However, this effect was only significant at the 10 percent level in the fractional probit model presented in Table 4.

According to H2, entry into the land rental market was negatively associated with trading labor for *ganyu* in seasonal agricultural labor markets. The recursive bivariate probit model results in Table 3 (model 4b) show that households that rent-in land are less likely to trade their labor for *ganyu*. If a farm household rented-in land, the probability of hiring out labor for *ganyu* statistically reduced by 38 percentage points at 1 percent significance level. These results are augmented by the observation that, at the 1 percent significance level, having no pre-rental land was positively associated with increasing land rentals by 5 percentage points and negatively associated with hiring out labor by 11 percentage points. Finally, I jointly assess H3 and H4 by comparing the statistically significant differences across the asset-wealth-to-labor-endowment-ratio quartiles, with the highest quartile as the reference group. Recall that H3 stated that increases to the household-asset-wealth-to-labor-endowment ratio increases entry to and extent of participation in the land rental markets for tenant household, while H4 stated that increases to the household-asset-wealth-to-labor-endowment ratio reduces entry to and extent of hiring out labor for casual agricultural work.

The bivariate and recursive bivariate model results in Table 3 show that households in the lower quartiles for asset-wealth-to-labor-endowment ratio were less likely to rent-in land compared to those in the upper quartiles (based on the negative sign in the tables). However, the negative association was only significant between the lowest and highest quartiles, potentially implying that households that were poor in terms of household-asset-wealth-to-labor-endowment ratio were more likely to be rationed out of land rental markets than intermediate groups. In the seasonal agricultural labor markets, a slightly different association was observed, with results indicating that the households across all of the lower quartiles were more likely to hire out labor than households in the highest quartile. That is, being in the 25th, 50th and 75th quartile increased the probability of hiring out labor compared to being in the above 75th quartile. The results in Table 4 show a similar effect of the asset-wealth-to-labor-endowment ratio on the extent of participation in both markets.

## 5. Discussion

This study investigated how changes in owned-land-to-labor-endowment ratios influence farm household-level decisions to either rent-in land or hire out labor among smallholders in Malawi. Based on the farm household decision model and empirical evidence, the results confirm this study's four hypotheses and substantially contribute to the literature on land scarcity relative to family labor and the trading of factors of production as livelihood strategies in SSA [5,9,23]. This study observed that a decreased land-to-labor ratio led households to trade in the agricultural land rental and seasonal agricultural labor markets [9,39]. Considering the growing land scarcity challenges and population pressure in SSA, this study's primary contribution is the confirmation of the likelihood of farm households increasing their participation in both markets.

In Malawi, studies separately assessing these markets have demonstrated a growing land rental market, even as more households supply labor to seasonal agricultural labor markets for short-term gains [19,21,34,58]. By jointly assessing the two markets, this study provides evidence that a unit decrease in the owned-land-to-labor-endowment ratio has double the effect on the casual labor market than on the land rental market. However, if the falling owned-land-to-labor-endowment ratio is a push factor for either the land rental or seasonal agricultural labor markets, it is important to understand how land rental market decisions influence farm household decisions to hire out family labor for *ganyu*.

Accordingly, this study's second contribution is recognizing that potential tenant households are more likely to reduce their use of family labor for *ganyu* if they rent-in land. This shift indicates how land rental markets can improve labor use on owned-land for medium- to long-term gains, an advantage compared to the short-term gains associated with *ganyu* [6,39]. Regarding *ganyu* as a coping strategy, it has been argued that Malawian households hire out labor for casual work because of the non-existence of land rental markets that could facilitate better use of land and labor by smallholders [19]. This paper's results concur, indicating that land rental markets can reduce the hiring out of labor for casual work among potential tenant households.

The observed negative association of land rental market decisions with casual labor market decisions is further augmented by the observation that having no pre-rental land was likely to push farm households to rent-in land rather than hire out agricultural labor. These results could imply that households are more likely to allocate land and labor for their own production—-enabling the achievement of food self-sufficiency objectives—-than rely on short-term wages from labor markets [19,21]. As land rental markets develop in SSA, this calls for the need to consider land rental markets as an affordable means of accessing agricultural land, which can allow for more profitable use of farm household labor by smallholders. However, it is worth questioning who is capable of renting-in land.

As discussed, casual work in seasonal agricultural labor markets is mostly associated with relative poverty, used as a strategy for overcoming idiosyncratic challenges, especially among female-headed households [10,15,16,58]. Contradicting this perception, this paper's third contribution suggests that not only the very poor, but the majority of the Malawian smallholders engage in *ganyu*. This corresponds with other findings that indicate that undertaking *ganyu* could be an important alternative source of income at the farm-household level [10,14,17]. Although this study did not focus on gender dynamics in the resource factor markets, the results presented in the appendix demonstrate that female-headed households were less likely to hire out labor for casual work, which differs from the previous literature. In contrast, this study concurs with literature that indicates that that female-headed households are less likely to be tenants in the land rental markets [20,59]. However, a detailed study is necessary to investigate gender dynamics and factor markets beyond female-headed households and the differences between patrilineal and matrilineal lineage systems.

The results are generally consistent with the extant literature, demonstrating that participation in land rental markets increases with increased capital (asset wealth) and labor endowment, with the very poor being rationed out [21]. However, hiring out labor for *ganyu* in seasonal agricultural labor markets is generally common among poorer households, including the majority of smallholders, due to their low capital relative to labor endowment [17]. This result indicates farm household capital or liquidity constraints could affect agricultural activities such as renting-in land and explains why farmers in Malawi might resort to short-term solutions such as *ganyu*.

## 6. Conclusions

Given increasing land scarcity challenges—caused by both population pressure and increased urbanization—and the continued development of land rental markets in most countries in SSA [5,9], this paper's objective was two-fold. First, I assessed the extent to which the falling owned-land-to-labor-endowment and asset-wealth-to-labor-endowment ratios were affecting farm household decisions to either rent-in agricultural land or hire out family labor as trade strategies in

response to growing land scarcity. Second, I assessed how land rental markets were influencing farm household decisions to hire out family labor for short-term casual agricultural work. Using data from Malawi Living Standards Measurement Surveys conducted in 2013 and 2016, I constructed a balanced household panel with which I combined district-level rainfall data to control for within-region rainfall-related production shocks. Furthermore, I used system approaches to jointly analyze decisions to rent-in land or hire out labor while controlling for possible endogeneity, simultaneity, and unobserved heterogeneity.

The results indicate that the falling owned-land-to-labor-endowment ratio is a push factor for farm households, encouraging participation in either land rental or seasonal agricultural labor markets. However, renting-in agricultural land can reduce entry into labor markets, where households engage in casual work to earn short-term wage. Meanwhile, according to the asset-wealth-to-labor-endowment ratio, wealthier farm households are more likely to rent-in land, with poorer households—in terms of asset-wealth-to-labor-endowment ratio—and the majority of smallholder households more likely to hire out labor for casual work.

The higher probability of smallholders hiring out labor for casual work could indicate household liquidity constraints related to agricultural operation costs and other household needs or higher friction or transaction costs in the land rental markets compared to the labor markets. Factor markets in SSA are mostly characterized by high transaction costs [43], while the land markets developing in SSA have been observed to feature high friction rates [9,57]. Accordingly, households could prefer to hire out labor rather than rent-in land due to these costs. To ensure that factor markets efficiently allocate land and labor, agricultural and land-use policies should focus on easing the liquidity burden of potential tenant households [60]. Such programs could include input subsidies or cash transfers at the beginning of the agricultural season. Regarding market friction, policy strategies such as land campaigns, access to land market information at the community level, and even establishing a land bank could facilitate access to capital for land rental transactions, particularly among younger generations [57]. Such resources could help farm households achieve food self-sufficiency and sustain their livelihoods. However, further research is needed regarding how the decisions of both potential tenants and landlords influence agricultural labor market decisions over time.

**Supplementary Materials:** The following are available online at http://www.mdpi.com/2073-445X/9/12/512/s1. Three STATA do files: (i) "IHPS data cleaning for land and labor each year" do file; (ii) "IHPS appended data for all years, labelling variables" do file; and (iii) "Balanced data for land and labor analysis" do file.

**Funding:** This research paper received no external funding.

**Acknowledgments:** This paper forms part of my Ph.D. research, undertaken through the Capacity Building for Climate Smart Natural Resource Management and Policy (CLISNARP) project. The CLISNARP project was supported by the NORAD-funded NORHED program from the Norwegian government, who had no role in this paper. I am grateful to my advisor, Stein T. Holden, for his valuable comments in conceptualizing the paper. I also acknowledge the support of the Norwegian University of Life Sciences (School of Economics and Business), Mekelle University in Ethiopia and the Lilongwe University of Agriculture and Natural Resources (LUANAR) in Malawi for facilitating the CLISNARP project. Finally, I appreciation the World Bank–LSMS team for the data, and I am grateful to the reviewers and editor for their valuable comments on this paper.

**Conflicts of Interest:** I declare no conflict of interest.

## Appendix A

**Table A1.** Summary of potential optimal household trade response strategies from the first order conditions (FOCs).

| | | Trade Response Strategies | | |
|---|---|---|---|---|
| | | **Labor (Equation (2))** | | |
| | | **Buyer/hiring in**<br>($L^i > 0$) | **Non-participant**<br>($L^i = 0 = L^o$) | **Seller/hiring out**<br>($L^o > 0$) |
| **Land (Equation (1))** | **Buyer/renting in ($A^i > 0$)** | Hiring in labor or renting-in land<br>Labor poor<br>1. $MR_{L^i} = MC_{L^i}$<br>Land poor<br>2. $MR_{A^i} = MC_{A^i}$ | Not trading labor but renting-in land<br>Labor sufficient<br>1. $MC_{L^o} < MR_A < MC_{L^i}$<br>Land poor<br>2. $MR_{A^i} = MC_{A^i}$ | Hiring out labor or renting-in the land<br>Labor rich<br>1. $MR_{L^o} = MC_{L^o}$<br>Land poor<br>2. $MR_{A^i} = MC_{A^i}$ |
| | **Non-participant**<br>( $A^o = 0$<br>$A^i = 0$ ) | Hiring in labor and not trading land<br>Labor poor<br>1. $MR_{L^i} = MC_{L^i}$<br>Land sufficient<br>2. $MR_{A^o} < MR_A < MC_{A^i}$ | Not trading labor and land<br>Labor sufficient<br>1. $MC_{L^o} < MR_{\overline{L}} < MC_{L^i}$<br>Land sufficient<br>2. $MR_{A^o} < MR_{\overline{A}} < MC_{A^i}$ | Hiring out labor and not trading land<br>Labor rich<br>1. $MR_{L^o} = MC_{L^o}$<br>Land sufficient<br>2. $MR_{A^o} < MR_{\overline{A}} < MC_{A^i}$ |
| | **Seller/renting out**<br>($A^o > 0$) | Hiring in labor or renting out the land<br>Labor poor<br>1. $MR_{L^i} = MC_{L^i}$<br>Land rich<br>2. $MR_{A^o} = MC_{A^o}$ | Not trading labor but renting out the land<br>Labor sufficient<br>1. $MC_{L^o} < MR_{\overline{L}} < MC_{L^i}$<br>Land rich<br>2. $MR_{A^o} = MC_{A^o}$ | Hiring out labor or renting out the land<br>Labor rich<br>1. $MR_{L^o} = MC_{L^o}$<br>Land rich<br>2. $MR_{A^o} = MC_{A^o}$ |

Note: MR is marginal revenue and MC is marginal cost with respect to land ($A$) and labor ($L$).

Factor component analysis variables for capital asset index:

**Durable assets**

(1) Mortar/pestle (mtondo); (2) bed; (3) table; (4) chair; (5) fan; (6) air conditioner; (7) radio ('wireless'); (8) tape or Compact Disk (CD) or Digital Versatile Disk (DVD) player or High-Fidelity (HiFi) sound system; (9) television; (10) Video Cassette Recorder (VCR); (11) sewing machine; (12) kerosene/paraffin stove; (13) electric or gas stove; (14) hot plate; (15) refrigerator; (16) washing machine; (17) bicycle; (18) motorcycle/scooter; (19) car; (20) mini-bus; (21) lorry; (22) beer-brewing drum; (23) upholstered chair; (24) sofa set; (25) coffee table (for the sitting room); (26) cupboard; (27) drawers; (28) bureau; (29) lantern (paraffin); (30) desk; (31) clock; (32) iron (for pressing clothes); (33) computer equipment and accessories; (34) satellite dish; (35) solar panel; (36) generator; (37) radio with flash drive/micro CD.

**Farm implements**

(1) Hand hoe; (2) slasher; (3) axe; (4) sprayer; (5) panga knife; (6) sickle; (7) treadle pump; (8) watering can; (9) ox cart; (10) ox plough; (11) tractor;

(12) tractor plough; (13) ridger; (14) cultivator; (15) motorized pump; (16) grain mill; (17) chicken house; (18) livestock kraal; (19) poultry kraal; (20) storage house; (21) granary; (22) pig sty.

**Table A2.** Bivariate probit model with conditional mixed process (CMP) margins for land rental and casual labor (*ganyu*) market participation.

| | Bivariate Probit | | Recursive Bivariate Probit | | Bivariate Probit | | Recursive Bivariate Probit | |
|---|---|---|---|---|---|---|---|---|
| | Random Effects (CMP Margins) | | Random Effects (CMP Margins) | | Correlated Random Effects (CMP Margins) | | Correlated Random Effects (CMP Margins) | |
| | 1a | 1b | 2a | 2b | 3a | 3b | 4a | 4b |
| VARIABLES | Rent in | Hire out | Rent in | Hire out | Rent in | Hire out | Rent in | Hire out |
| **Key variables** | | | | | | | | |
| Land rented in (1 = Yes) | | | | −0.379 **** | | | | −0.375 **** |
| | | | | (0.09) | | | | (0.09) |
| Owned-farmland-to-labor ratio | −0.095 ** | −0.202 **** | −0.101 *** | −0.222 **** | −0.096 ** | −0.199 **** | −0.102 *** | −0.219 **** |
| (ha/adult equiv. labor unit) | (0.04) | (0.04) | (0.04) | (0.04) | (0.04) | (0.04) | (0.04) | (0.04) |
| Asset-wealth-index-to-labor ratio | | | | | | | | |
| Base: Quartile 4 | | | | | | | | |
| Quartile 1 | −0.037 ** | 0.236 **** | −0.045 *** | 0.209 **** | −0.038 ** | 0.213 **** | −0.046 *** | 0.188 **** |
| | (0.02) | (0.02) | (0.02) | (0.03) | (0.02) | (0.03) | (0.02) | (0.03) |
| Quartile 2 | −0.005 | 0.275 **** | −0.010 | 0.254 **** | −0.007 | 0.253 **** | −0.011 | 0.235 **** |
| | (0.02) | (0.02) | (0.02) | (0.02) | (0.02) | (0.02) | (0.02) | (0.02) |
| Quartile 3 | 0.021 | 0.196 **** | 0.019 | 0.190 **** | 0.021 | 0.178 **** | 0.019 | 0.174 **** |
| | (0.01) | (0.02) | (0.01) | (0.02) | (0.02) | (0.02) | (0.01) | (0.02) |
| No pre-rental land (1 = yes) | 0.052 **** | −0.138 **** | 0.047 *** | −0.106 **** | 0.053 **** | −0.132 **** | 0.049 **** | −0.100 **** |
| | (0.01) | (0.02) | (0.01) | (0.02) | (0.02) | (0.02) | (0.01) | (0.02) |
| **Rainfall variations** | | | | | | | | |
| **Positive** deviation (dm) **one-year lag** | 0.001 | −0.007 | −0.000 | −0.006 | 0.001 | −0.006 | −0.000 | −0.006 |
| (Early plus mid-season) | (0.00) | (0.01) | (0.00) | (0.01) | (0.00) | (0.01) | (0.00) | (0.01) |
| **Absolute negative** deviation (dm) **one-year lag** | 0.020 *** | −0.028 *** | 0.020 *** | −0.018 * | 0.020 *** | −0.027 ** | 0.020 *** | −0.017 |
| (Early plus mid-season) | (0.01) | (0.01) | (0.01) | (0.01) | (0.01) | (0.01) | (0.01) | (0.01) |
| **Farm and household characteristics** | | | | | | | | |
| **Observed control variables** | | | | | | | | |
| Sex of household head (HH) (1 = Female) | −0.040 *** | −0.042 ** | −0.039 *** | −0.050 *** | | | | |
| | (0.01) | (0.02) | (0.01) | (0.02) | | | | |
| Age of HH (years) | −0.001 * | −0.004 **** | −0.001 ** | −0.004 **** | | | | |
| | (0.00) | (0.00) | (0.00) | (0.00) | | | | |
| Education of HH (years) | 0.001 | −0.014 **** | 0.001 | −0.013 **** | | | | |
| | (0.00) | (0.00) | (0.00) | (0.00) | | | | |
| Household-size-to-labor ratio | 0.024 ** | −0.003 | 0.025 *** | 0.008 | | | | |
| (No. of persons/adult equiv. labor unit) | (0.01) | (0.02) | (0.01) | (0.02) | | | | |

**Table A2.** *Cont.*

| VARIABLES | Bivariate Probit Random Effects (CMP Margins) | | Recursive Bivariate Probit Random Effects (CMP Margins) | | Bivariate Probit Correlated Random Effects (CMP Margins) | | Recursive Bivariate Probit Correlated Random Effects (CMP Margins) | |
|---|---|---|---|---|---|---|---|---|
| | 1a | 1b | 2a | 2b | 3a | 3b | 4a | 4b |
| | Rent in | Hire out | Rent in | Hire out | Rent in | Hire out | Rent in | Hire out |
| Total Livestock Units (TLU)-to-labor ratio | 0.007 | −0.016 | 0.006 | −0.011 | | | | |
| | (0.01) | (0.02) | (0.00) | (0.02) | | | | |
| One-year lag TLU-to-labor ratio | 0.004 | −0.054 | 0.003 | −0.046 | | | | |
| | (0.00) | (0.03) | (0.00) | (0.03) | | | | |
| Distance to the nearest city zone (km) | 0.002 **** | 0.001 | 0.002 **** | 0.001 *** | | | | |
| | (0.00) | (0.00) | (0.00) | (0.00) | | | | |
| Mean of observed control variables | | | | | | | | |
| Sex of HH (1 = Female) | | | | | −0.040 *** | −0.032 | −0.038 *** | −0.041 ** |
| | | | | | (0.01) | (0.02) | (0.01) | (0.02) |
| Age of HH (years) | | | | | −0.001 | −0.004 **** | −0.001 * | −0.004 **** |
| | | | | | (0.00) | (0.00) | (0.00) | (0.00) |
| Education of HH (years) | | | | | 0.001 | −0.018 **** | 0.001 | −0.016 **** |
| | | | | | (0.00) | (0.00) | (0.00) | (0.00) |
| Household-size-to-labor ratio | | | | | 0.035 ** | 0.009 | 0.033 ** | 0.021 |
| (No. of persons/adult equiv. labor unit) | | | | | (0.01) | (0.02) | (0.01) | (0.02) |
| Total Livestock Units (TLU)-to-labor ratio | | | | | 0.009 | −0.016 | 0.007 | −0.010 |
| | | | | | (0.01) | (0.02) | (0.01) | (0.02) |
| One-year lag TLU-to-labor ratio | | | | | 0.004 | −0.066 | 0.004 | −0.055 |
| | | | | | (0.01) | (0.04) | (0.01) | (0.04) |
| Distance to the nearest city zone (km) | | | | | 0.002 **** | 0.001 | 0.002 **** | 0.001 *** |
| | | | | | (0.00) | (0.00) | (0.00) | (0.00) |
| **Deviations from the mean** | | | | | | | | |
| Sex of HH (1 = Female) | | | | | −0.037 | −0.083 * | −0.031 | −0.088 * |
| | | | | | (0.02) | (0.05) | (0.02) | (0.05) |
| Age of HH (years) | | | | | −0.000 | −0.001 | −0.000 | −0.001 |
| | | | | | (0.00) | (0.00) | (0.00) | (0.00) |
| Education of HH (years) | | | | | −0.001 | 0.009 ** | −0.000 | 0.009 ** |
| | | | | | (0.00) | (0.00) | (0.00) | (0.00) |
| Household-size-to-labor ratio | | | | | 0.009 | −0.030 | 0.011 | −0.023 |
| (No. of persons/adult equiv. labor) | | | | | (0.01) | (0.02) | (0.01) | (0.02) |

**Table A2.** *Cont.*

| VARIABLES | Bivariate Probit Random Effects (CMP Margins) | | Recursive Bivariate Probit Random Effects (CMP Margins) | | Bivariate Probit Correlated Random Effects (CMP Margins) | | Recursive Bivariate Probit Correlated Random Effects (CMP Margins) | |
|---|---|---|---|---|---|---|---|---|
| | 1a | 1b | 2a | 2b | 3a | 3b | 4a | 4b |
| | Rent in | Hire out | Rent in | Hire out | Rent in | Hire out | Rent in | Hire out |
| Total Livestock Units (TLU)-to-labor ratio | | | | | −0.000 | −0.044 | −0.000 | −0.040 |
| | | | | | (0.01) | (0.04) | (0.01) | (0.04) |
| One-year lag TLU-to-labor ratio | | | | | 0.004 | −0.019 | 0.003 | −0.017 |
| | | | | | (0.00) | (0.05) | (0.00) | (0.04) |
| Distance to the nearest city zone (km) | | | | | 0.001 | 0.000 | 0.001 | 0.001 |
| | | | | | (0.00) | (0.00) | (0.00) | (0.00) |
| **Regional dummy (1 = Central)** | | | | | | | | |
| 2. Northern region | −0.127 **** | 0.015 | −0.125 **** | −0.036 | −0.128 **** | 0.017 | −0.127 **** | −0.034 |
| | (0.01) | (0.03) | (0.01) | (0.03) | (0.01) | (0.03) | (0.01) | (0.03) |
| 3. Southern region | −0.073 **** | 0.001 | −0.072 **** | −0.023 | −0.074 **** | −0.003 | −0.073 **** | −0.027 |
| | (0.01) | (0.02) | (0.01) | (0.02) | (0.01) | (0.02) | (0.01) | (0.02) |
| 2016.year | −0.016 * | 0.223 **** | −0.012 | 0.202 **** | −0.017 * | 0.218 **** | −0.015 | 0.198 **** |
| | (0.01) | (0.02) | (0.01) | (0.02) | (0.01) | (0.02) | (0.01) | (0.02) |
| N | 3790 | 3790 | 3790 | 3790 | 3790 | 3790 | 3790 | 3790 |

Note: standard errors in parentheses. **** indicates $p < 0.001$; *** indicates $p < 0.01$; ** indicates $p < 0.05$; * indicates $p < 0.1$.

**Table A3.** Coefficients of the bivariate probit model with conditional mixed process (CMP) for land rental and casual labor (*ganyu*) market participation.

| | Bivariate Probit | | Recursive Bivariate Probit | | Bivariate Probit | | Recursive Bivariate Probit | |
|---|---|---|---|---|---|---|---|---|
| | Random Effects (CMP Coefficients) | | Random Effects (CMP Coefficients) | | Correlated Random Effects (CMP Coefficients) | | Correlated Random Effects (CMP Coefficients) | |
| | 1a | 1b | 2a | 2b | 3a | 3b | 4a | 4b |
| **VARIABLES** | Rent in | Hire out | Rent in | Hire out | Rent in | Hire out | Rent in | Hire out |
| **Key variables** | | | | | | | | |
| Land rented in (1 = Yes) | | | | −1.140 **** | | | | −1.134 **** |
| | | | | (0.28) | | | | (0.29) |
| Owned-farmland-to-labor ratio | −0.633 ** | −0.594 **** | −0.674 *** | −0.667 **** | −0.638 ** | −0.589 **** | −0.680 *** | −0.664 **** |
| (ha/adult equiv. labor unit) | (0.27) | (0.12) | (0.23) | (0.13) | (0.27) | (0.12) | (0.24) | (0.13) |
| Asset-wealth-index-to-labor ratio | | | | | | | | |
| Base: Quartile 4 | | | | | | | | |
| Quartile 1 | −0.247 ** | 0.696 **** | −0.300 *** | 0.630 **** | −0.251 ** | 0.632 **** | −0.306 *** | 0.571 **** |
| | (0.11) | (0.08) | (0.11) | (0.08) | (0.12) | (0.08) | (0.12) | (0.08) |
| Quartile 2 | −0.035 | 0.809 **** | −0.064 | 0.766 **** | −0.044 | 0.752 **** | −0.072 | 0.712 **** |
| | (0.11) | (0.07) | (0.10) | (0.08) | (0.11) | (0.07) | (0.11) | (0.08) |
| Quartile 3 | 0.142 | 0.577 **** | 0.126 | 0.573 **** | 0.140 | 0.529 **** | 0.124 | 0.527 **** |
| | (0.10) | (0.07) | (0.09) | (0.07) | (0.10) | (0.07) | (0.10) | (0.07) |
| No pre-rental land (1 = yes) | 0.347 **** | −0.406 **** | 0.316 *** | −0.318 **** | 0.353 **** | −0.390 **** | 0.324 *** | −0.304 **** |
| | (0.10) | (0.06) | (0.10) | (0.07) | (0.10) | (0.06) | (0.10) | (0.07) |
| **Rainfall variations** | | | | | | | | |
| **Positive** deviation (dm) **one-year lag** | 0.004 | −0.020 | −0.003 | −0.019 | 0.003 | −0.018 | −0.003 | −0.017 |
| (Early plus mid-season) | (0.02) | (0.02) | (0.02) | (0.02) | (0.02) | (0.02) | (0.02) | (0.02) |
| **Absolute negative** deviation (dm) **one-year lag** | 0.132 *** | −0.084 *** | 0.131 *** | −0.054 * | 0.133 *** | −0.082 ** | 0.131 *** | −0.052 |
| (Early plus mid-season) | (0.04) | (0.03) | (0.04) | (0.03) | (0.04) | (0.03) | (0.04) | (0.03) |
| **Farm and household characteristics** | | | | | | | | |
| **Observed control variables** | | | | | | | | |
| Sex of household head (HH) (1 = Female) | −0.268 *** | −0.123 ** | −0.257 *** | −0.150 *** | | | | |
| | (0.09) | (0.06) | (0.09) | (0.06) | | | | |
| Age of HH (years) | −0.005 * | −0.011 **** | −0.006 ** | −0.011 **** | | | | |
| | (0.00) | (0.00) | (0.00) | (0.00) | | | | |
| Education of HH (years) | 0.004 | −0.041 **** | 0.004 | −0.038 **** | | | | |
| | (0.01) | (0.01) | (0.01) | (0.01) | | | | |
| Household-size-to-labor ratio | 0.163 ** | −0.008 | 0.165 *** | 0.023 | | | | |
| (No. of persons/adult equiv. labor unit) | (0.06) | (0.05) | (0.06) | (0.05) | | | | |

**Table A3.** *Cont.*

| VARIABLES | Bivariate Probit Random Effects (CMP Coefficients) | | Recursive Bivariate Probit Random Effects (CMP Coefficients) | | Bivariate Probit Correlated Random Effects (CMP Coefficients) | | Recursive Bivariate Probit Correlated Random Effects (CMP Coefficients) | |
|---|---|---|---|---|---|---|---|---|
| | 1a | 1b | 2a | 2b | 3a | 3b | 4a | 4b |
| | Rent in | Hire out | Rent in | Hire out | Rent in | Hire out | Rent in | Hire out |
| Total Livestock Units (TLU)-to-labor ratio | 0.046 | −0.047 | 0.042 | −0.034 | | | | |
| | (0.04) | (0.07) | (0.03) | (0.05) | | | | |
| One-year lag TLU-to-labor ratio | 0.027 | −0.159 | 0.023 | −0.140 | | | | |
| | (0.03) | (0.10) | (0.03) | (0.09) | | | | |
| Distance to the nearest city zone (km) | 0.015 **** | 0.002 | 0.015 **** | 0.004 *** | | | | |
| | (0.00) | (0.00) | (0.00) | (0.00) | | | | |
| **Mean of observed control variables** | | | | | | | | |
| Sex of HH (1 = Female) | | | | | −0.268 *** | −0.095 | −0.256 *** | −0.123 ** |
| | | | | | (0.10) | (0.06) | (0.10) | (0.06) |
| Age of HH (years) | | | | | −0.004 | −0.012 **** | −0.006 * | −0.012 **** |
| | | | | | (0.00) | (0.00) | (0.00) | (0.00) |
| Education of HH (years) | | | | | 0.005 | −0.053 **** | 0.006 | −0.050 **** |
| | | | | | (0.01) | (0.01) | (0.01) | (0.01) |
| Household-size-to-labor ratio | | | | | 0.233 ** | 0.026 | 0.223 ** | 0.064 |
| (No. of persons/adult equiv. labor unit) | | | | | (0.09) | (0.06) | (0.09) | (0.06) |
| Total Livestock Units (TLU)-to-labor ratio | | | | | 0.057 | −0.047 | 0.050 | −0.031 |
| | | | | | (0.05) | (0.07) | (0.04) | (0.05) |
| One-year lag TLU-to-labor ratio | | | | | 0.030 | −0.197 | 0.025 | −0.167 |
| | | | | | (0.05) | (0.12) | (0.05) | (0.11) |
| Distance to the nearest city zone (km) | | | | | 0.015 **** | 0.002 | 0.015 **** | 0.004 *** |
| | | | | | (0.00) | (0.00) | (0.00) | (0.00) |
| **Deviations from the mean** | | | | | | | | |
| Sex of HH (1 = Female) | | | | | −0.250 | −0.247 * | −0.210 | −0.267 * |
| | | | | | (0.16) | (0.15) | (0.15) | (0.15) |
| Age of HH (years) | | | | | −0.003 | −0.002 | −0.001 | −0.003 |
| | | | | | (0.01) | (0.01) | (0.01) | (0.01) |
| Education of HH (years) | | | | | −0.003 | 0.028 ** | −0.002 | 0.026 ** |
| | | | | | (0.02) | (0.01) | (0.02) | (0.01) |

**Table A3.** *Cont.*

| VARIABLES | Bivariate Probit Random Effects (CMP Coefficients) | | Recursive Bivariate Probit Random Effects (CMP Coefficients) | | Bivariate Probit Correlated Random Effects (CMP Coefficients) | | Recursive Bivariate Probit Correlated Random Effects (CMP Coefficients) | |
|---|---|---|---|---|---|---|---|---|
| | 1a | 1b | 2a | 2b | 3a | 3b | 4a | 4b |
| | Rent in | Hire out | Rent in | Hire out | Rent in | Hire out | Rent in | Hire out |
| Household-size-to-labor ratio | | | | | 0.063 | −0.089 | 0.073 | −0.071 |
| (No. of persons/adult equiv. labor unit) | | | | | (0.05) | (0.07) | (0.05) | (0.07) |
| Total Livestock Units (TLU)-to-labor ratio | | | | | −0.002 | −0.129 | −0.001 | −0.121 |
| | | | | | (0.03) | (0.13) | (0.04) | (0.12) |
| One-year lag TLU-to-labor ratio | | | | | 0.024 | −0.056 | 0.021 | −0.051 |
| | | | | | (0.02) | (0.15) | (0.02) | (0.13) |
| Distance to the nearest city zone (km) | | | | | 0.005 | 0.001 | 0.003 | 0.002 |
| | | | | | (0.00) | (0.00) | (0.00) | (0.00) |
| **Regional dummy (1 = Central)** | | | | | | | | |
| 2. Northern region | −1.168 **** | 0.045 | −1.143 **** | −0.109 | −1.181 **** | 0.050 | −1.155 **** | −0.103 |
| | (0.15) | (0.08) | (0.14) | (0.09) | (0.15) | (0.08) | (0.14) | (0.09) |
| 3. Southern region | −0.441 **** | 0.004 | −0.436 **** | −0.069 | −0.448 **** | −0.008 | −0.445 **** | −0.081 |
| | (0.09) | (0.06) | (0.09) | (0.06) | (0.09) | (0.06) | (0.09) | (0.06) |
| 2016.year | −0.106 * | 0.645 **** | −0.081 | 0.600 **** | −0.116 * | 0.634 **** | −0.103 | 0.590 **** |
| | (0.06) | (0.05) | (0.06) | (0.05) | (0.06) | (0.05) | (0.07) | (0.05) |
| Constant | −1.633 **** | 0.258 | −1.541 **** | 0.281 | −1.808 **** | 0.375 * | −1.679 **** | 0.369 * |
| | (0.25) | (0.17) | (0.25) | (0.17) | (0.30) | (0.20) | (0.30) | (0.20) |
| atanhrho_12 | | −0.038 | | 0.625 *** | | −0.038 | | 0.620 *** |
| | | (0.04) | | (0.21) | | (0.04) | | (0.22) |
| Log pseudolikelihood | | −3308.3 | | −3304.3 | | −3289.7 | | −3285.8 |
| Observations | 3790 | 3790 | 3790 | 3790 | 3790 | 3790 | 3790 | 3,790 |

Note: standard errors in parentheses. **** indicates $p < 0.001$; *** indicates $p < 0.01$; ** indicates $p < 0.05$; * indicates $p < 0.1$.

**Table A4.** Conditional mixed process (CMP) margins from a systems approach using Tobit for the land rental markets and fractional probit for the share of adult equivalent labor hired out for casual work (*ganyu*).

| VARIABLES | Random Effects (CMP Margins) | | Correlated Random Effects (CMP Margins) | |
|---|---|---|---|---|
| | Tobit: Rent in | Fractional Probit: Hire out | Tobit: Rent in | Fractional Probit: Hire out |
| **Key variables** | | | | |
| Owned-farmland-to-labor ratio | −0.087 ** | −0.052 ** | −0.088 ** | −0.049 * |
| (ha/adult equiv. labor unit) | (0.04) | (0.03) | (0.04) | (0.03) |
| Asset-wealth-index-to-labor ratio. | | | | |
| Base: Quartile 4 | | | | |
| Quartile 1 | −0.047 *** | 0.243 **** | −0.047 *** | 0.229 **** |
| | (0.02) | (0.02) | (0.02) | (0.02) |
| Quartile 2 | −0.014 | 0.185 **** | −0.014 | 0.173 **** |
| | (0.02) | (0.02) | (0.02) | (0.02) |
| Quartile 3 | 0.017 | 0.105 **** | 0.018 | 0.095 **** |
| | (0.01) | (0.02) | (0.02) | (0.02) |
| No pre-rental land (1 = yes) | 0.054 **** | −0.070 **** | 0.055 **** | −0.067 **** |
| | (0.01) | (0.01) | (0.01) | (0.01) |
| **Rainfall variations** | | | | |
| **Positive** deviation (dm) **one-year lag** | −0.001 | −0.001 | −0.001 | −0.000 |
| (Early plus mid-season) | (0.00) | (0.00) | (0.00) | (0.00) |
| **Absolute negative** deviation (dm) **one-year lag** | 0.018 *** | −0.013 * | 0.018 *** | −0.013 * |
| (Early plus mid-season) | (0.01) | (0.01) | (0.01) | (0.01) |
| **Farm and household characteristics** | | | | |
| **Observed control variables** | | | | |
| Sex of household head (HH) (1 = Female) | −0.045 **** | −0.030 ** | | |
| | (0.01) | (0.01) | | |
| Age of HH (years) | −0.001 | −0.003 **** | | |
| | (0.00) | (0.00) | | |
| Education of HH (years) | 0.001 | −0.010 **** | | |
| | (0.00) | (0.00) | | |
| Household-size-to-labor ratio | 0.026 *** | −0.011 | | |
| (No. of persons/adult equiv. labor unit) | (0.01) | (0.01) | | |
| Total Livestock Units (TLU)-to-labor ratio | 0.007 | −0.026 | | |
| | (0.01) | (0.03) | | |

**Table A4.** *Cont.*

| VARIABLES | Random Effects (CMP Margins) | | Correlated Random Effects (CMP Margins) | |
|---|---|---|---|---|
| | Tobit: Rent in | Fractional Probit: Hire out | Tobit: Rent in | Fractional Probit: Hire out |
| One-year lag TLU-to-labor ratio | 0.005 | −0.022 | | |
| | (0.00) | (0.02) | | |
| Distance to the nearest city zone (km) | 0.002 **** | 0.001 ** | | |
| | (0.00) | (0.00) | | |
| **Mean of observed control variables** | | | | |
| Sex of HH (1 = Female) | | | −0.044 *** | −0.018 |
| | | | (0.01) | (0.01) |
| Age of HH (years) | | | −0.000 | −0.004 **** |
| | | | (0.00) | (0.00) |
| Education of HH (years) | | | 0.001 | −0.012 **** |
| | | | (0.00) | (0.00) |
| Household-size-to-labor ratio | | | 0.037 *** | −0.017 |
| (No. of persons/adult equiv. labor unit) | | | (0.01) | (0.02) |
| Total Livestock Units (TLU)-to-labor ratio | | | 0.009 | −0.024 |
| | | | (0.01) | (0.03) |
| One-year lag TLU-to-labor ratio | | | 0.005 | −0.041 |
| | | | (0.01) | (0.03) |
| Distance to the nearest city zone (km) | | | 0.002 **** | 0.001 ** |
| | | | (0.00) | (0.00) |
| **Deviations from the mean** | | | | |
| Sex of HH (1 = Female) | | | −0.044 * | −0.080 ** |
| | | | (0.02) | (0.03) |
| Age of HH (years) | | | −0.000 | 0.000 |
| | | | (0.00) | (0.00) |
| Education of HH (years) | | | −0.002 | 0.004 |
| | | | (0.00) | (0.00) |
| Household-size-to-labor ratio | | | 0.009 | −0.005 |
| (No. of persons/adult equiv. labor unit) | | | (0.01) | (0.02) |
| Total Livestock Units (TLU)-to-labor ratio | | | −0.001 | −0.038 |
| | | | (0.01) | (0.04) |

**Table A4.** *Cont.*

| | Random Effects (CMP Margins) | | Correlated Random Effects (CMP Margins) | |
|---|---|---|---|---|
| **VARIABLES** | **Tobit: Rent in** | **Fractional Probit: Hire out** | **Tobit: Rent in** | **Fractional Probit: Hire out** |
| One-year lag TLU-to-labor ratio | | | 0.004 | 0.011 |
| | | | (0.00) | (0.03) |
| Distance to the nearest city zone (km) | | | 0.001 | 0.001 |
| | | | (0.00) | (0.00) |
| **Regional dummy (1 = Central)** | | | | |
| 2. Northern region | −0.129 **** | −0.004 | −0.131 **** | −0.001 |
| | (0.01) | (0.02) | (0.01) | (0.02) |
| 3. Southern region | −0.066 **** | −0.007 | −0.068 **** | −0.010 |
| | (0.01) | (0.01) | (0.01) | (0.01) |
| 2016.year | −0.011 | 0.108 **** | −0.013 | 0.102 **** |
| | (0.01) | (0.01) | (0.01) | (0.01) |
| N | 3790 | 3790 | 3790 | 3790 |

Note: standard errors in parentheses. **** indicates $p < 0.001$; *** indicates $p < 0.01$; ** indicates $p < 0.05$; * indicates $p < 0.1$.

**Table A5.** Conditional mixed process (CMP) coefficients from a systems approach using Tobit for the land rental markets and fractional probit for the share of adult equivalent labor hired out for casual work (*ganyu*).

| | Random Effects (CMP Margins) | | Correlated Random Effects (CMP Margins) | |
|---|---|---|---|---|
| **VARIABLES** | **Tobit: Rent in** | **Fractional Probit: Hire out** | **Tobit: Rent in** | **Fractional Probit: Hire out** |
| **Key variables** | | | | |
| Owned-farmland-to-labor ratio | −0.549 ** | −0.172 ** | −0.552 ** | −0.164 * |
| (ha/adult equiv. labor unit) | (0.23) | (0.09) | (0.23) | (0.09) |
| Asset-wealth-index-to-labor ratio | | | | |
| Base: Quartile 4 | | | | |
| Quartile 1 | −0.299 *** | 0.806 **** | −0.297 *** | 0.764 **** |
| | (0.11) | (0.06) | (0.11) | (0.06) |
| Quartile 2 | −0.087 | 0.613 **** | −0.091 | 0.575 **** |
| | (0.10) | (0.06) | (0.10) | (0.06) |
| Quartile 3 | 0.109 | 0.347 **** | 0.112 | 0.316 **** |
| | (0.09) | (0.06) | (0.10) | (0.06) |
| No pre-rental land (1 = yes) | 0.342 **** | −0.232 **** | 0.346 **** | −0.222 **** |
| | (0.09) | (0.05) | (0.09) | (0.05) |

**Table A5.** *Cont.*

| VARIABLES | Random Effects (CMP Margins) | | Correlated Random Effects (CMP Margins) | |
|---|---|---|---|---|
| | Tobit: Rent in | Fractional Probit: Hire out | Tobit: Rent in | Fractional Probit: Hire out |
| **Rainfall variations** | | | | |
| **Positive** deviation (dm) **one-year lag** | −0.007 | −0.002 | −0.007 | −0.001 |
| (Early plus mid-season) | (0.02) | (0.01) | (0.02) | (0.01) |
| **Absolute negative** deviation (dm) **one-year lag** | 0.111 *** | −0.043 * | 0.110 *** | −0.043 * |
| (Early plus mid-season) | (0.04) | (0.02) | (0.04) | (0.02) |
| **Farm and household characteristics** | | | | |
| **Observed control variables** | −0.281 **** | −0.100 ** | | |
| Sex of household head (HH) (1 = Female) | (0.08) | (0.04) | | |
| | −0.004 | −0.011 **** | | |
| Age of HH (years) | (0.00) | (0.00) | | |
| | 0.004 | −0.032 **** | | |
| Education of HH (years) | (0.01) | (0.00) | | |
| | 0.162 *** | −0.035 | | |
| Household-size-to-labor ratio | (0.06) | (0.04) | | |
| (No. of persons/adult equiv. labor unit) | 0.044 | −0.087 | | |
| Total Livestock Units (TLU)-to-labor ratio | (0.04) | (0.10) | | |
| | 0.029 | −0.072 | | |
| One-year lag TLU-to-labor ratio | (0.03) | (0.08) | | |
| | 0.015 **** | 0.002 ** | | |
| Distance to the nearest city zone (km) | (0.00) | (0.00) | | |
| **Mean of observed control variables** | | | | |
| Sex of HH (1 = Female) | | | −0.277 *** | −0.061 |
| | | | (0.09) | (0.04) |
| Age of HH (years) | | | −0.003 | −0.013 **** |
| | | | (0.00) | (0.00) |
| Education of HH (years) | | | 0.007 | −0.040 **** |
| | | | (0.01) | (0.01) |
| Household-size-to-labor ratio | | | 0.234 *** | −0.056 |
| (No. of persons/adult equiv. labor unit) | | | (0.09) | (0.05) |
| Total Livestock Units (TLU)-to-labor ratio | | | 0.056 | −0.079 |
| | | | (0.05) | (0.09) |
| One-year lag TLU-to-labor ratio | | | 0.032 | −0.136 |
| | | | (0.05) | (0.10) |

**Table A5.** *Cont.*

| | Random Effects (CMP Margins) | | Correlated Random Effects (CMP Margins) | |
|---|---|---|---|---|
| **VARIABLES** | **Tobit: Rent in** | **Fractional Probit: Hire out** | **Tobit: Rent in** | **Fractional Probit: Hire out** |
| Distance to the nearest city zone (km) | | | 0.016 **** | 0.002 ** |
| | | | (0.00) | (0.00) |
| **Deviations from the mean** | | | | |
| Sex of HH (1 = Female) | | | −0.275 * | −0.265 ** |
| | | | (0.15) | (0.10) |
| Age of HH (years) | | | −0.001 | 0.000 |
| | | | (0.01) | (0.00) |
| Education of HH (years) | | | −0.010 | 0.013 |
| | | | (0.02) | (0.01) |
| Household-size-to-labor ratio | | | 0.058 | −0.016 |
| (No. of persons/adult equiv. labor unit) | | | (0.04) | (0.06) |
| Total Livestock Units (TLU)-to-labor ratio | | | −0.006 | −0.128 |
| | | | (0.04) | (0.12) |
| One-year lag TLU-to-labor ratio | | | 0.025 | 0.036 |
| | | | (0.02) | (0.10) |
| Distance to the nearest city zone (km) | | | 0.005 | 0.002 |
| | | | (0.00) | (0.00) |
| **Regional dummy (1 = Central)** | | | | |
| 2. Northern region | −1.145 **** | −0.014 | −1.156 **** | −0.005 |
| | (0.16) | (0.06) | (0.16) | (0.06) |
| 3. Southern region | −0.380 **** | −0.024 | −0.387 **** | −0.032 |
| | (0.08) | (0.04) | (0.08) | (0.04) |
| 2016.year | −0.072 | 0.358 **** | −0.084 | 0.338 **** |
| | (0.06) | (0.03) | (0.06) | (0.04) |
| Constant | −1.614 **** | −0.421 *** | −1.802 **** | −0.230 |
| | (0.25) | (0.13) | (0.31) | (0.16) |
| lnsig_1 | | −0.040 | | −0.043 |
| | | (0.06) | | (0.06) |
| atanhrho_12 | | −0.018 | | −0.018 |
| | | (0.03) | | (0.03) |
| Log pseudolikelihood | | −3160.5 | | −3160.5 |
| Observations | 3790 | 3790 | 3790 | 3790 |

Note: standard errors in parentheses. **** indicates $p < 0.001$; *** indicates $p < 0.01$; ** indicates $p < 0.05$; * indicates $p < 0.1$.

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
