# Peer review of "Agricultural Resources and Trade Strategies: Response to Falling Land-to-Labor Ratios in Malawi"

_land, doi:10.3390/land9120512_

Round 1

Reviewer 1 Report

The topic discussed in this paper tackles some fascinating issues involving the relationship between land scarcity relative to family labor.
Such a contribution could be of high interest in Land journal. I would like to see this paper in a revised form.
In particular, in my personal view and its current state, the paper suffers some limitations that should be faced and resolved. The paper’s presentational style is low, with some unintelligible passages. Moreover, an extension of the theory should be useful. Comparative statics seems to be not so helpful in this case. Am I missing something? Furthermore, I do not see the perfect link with the empirical analysis. Please try to explain better. Moreover, a good comparision with the literature should be useful (the background section is not enough).

I thus recommend a revision of the writing and the paper’s presentation, focusing on the elements specified above (these should be fulfilled for publication to be endorsed).

Author Response

General comment

The topic discussed in this paper tackles some fascinating issues involving the relationship between land scarcity relative to family labor. Such a contribution could be of high interest in the Land journal. I would like to see this paper in a revised form. In particular, in my personal view and its current state, the paper suffers some limitations that should be faced and resolved.

General response

Thank you for taking the time to review the paper and for giving me a chance to revise and resubmit.

After carefully considering all the comments from reviewers, I have improved the title as the highlight of the study. The new title is “Agricultural resources and trade strategies: Response to falling land-to-labor ratios in Malawi”.

Question 1

The paper’s presentational style is low, with some unintelligible passages.

Response 1

I have re-organized the paper by having separate sections on results and discussion. I have also renamed the “Materials and Methods” section to the “Data and Estimation Method” section. By re-organizing, I have discussed the results further while referring to extant literature. Overall, I also solicited help on editing the English from Cambridge proofreading to improve my paper.  

Question 2

Moreover, an extension of the theory should be useful. Comparative statics seems to be not so helpful in this case. Am I missing something?

Response 2

In the theoretical framework and hypotheses section, I have extended the farm household model to show second-order conditions, Hessian matrix and comparative statics for renting in and hiring out labour in line with the paper’s objective.

Question 3

Furthermore, I do not see the perfect link with the empirical analysis. Please try to explain better.

Response 3

It is from the second-order conditions and the comparative statics that I draw the direction of the effect in hypotheses for empirical analysis.  

Question 4

Moreover, a good comparison with the literature should be useful (the background section is not enough). I thus recommend a revision of the writing and the paper’s presentation, focusing on the elements specified above (these should be fulfilled for publication to be endorsed).

Response 4

In the background section, I have added information on land distribution, holding sizes, classification of small- and large-scale farms, and governance systems in Malawi. Additionally, by having a separate section on the discussion of results, I reviewed and compared my finding with literature.

I hope these changes fulfils a consideration in the journal,

Thank you.

Reviewer 2 Report

Thank you for an interesting study on the household decision related to livelihood strategies in response to land and family labour relations.The article I have read with great interest, a bit difficult to understand is the title, it could be written in a more in a clear and transparent way, to exposing the household strategy as a basis of reasoning, (i.e. start with) the part referring to the livelihood strategy in response to changes in the land and market. Now it is hard to read as good highlight of the study.
The article itself is written with great care and in accordance with the principles of the research approach used in studies of this type, which is further confirmed by the analysis of the attached, supplemnetary files.
In the results or conclusions section could one be tempted to make a additional paragraph discussing the results obtained from the research with those available in the literature of the subject. However, the main results were supported by literature references, and overall reasoning of the study and conclusions are correct.

Thank you!

Author Response

General Comment

Thank you for an interesting study on the household decision related to livelihood strategies in response to land and family labor relations. The article itself is written with great care and in accordance with the principles of the research approach used in studies of this type, which is further confirmed by the analysis of the attached, supplementary files.

General response

Thank you for taking the time to review the paper and for giving me a chance to revise and resubmit.

After carefully considering all the comments from reviewers, I have improved the title as the highlight of the study. The new title is “Agricultural resources and trade strategies: Response to falling land-to-labor ratios in Malawi”.

Question 1

The article I have read with great interest, a bit difficult to understand is the title, it could be written in a more in a clear and transparent way, to exposing the household strategy as a basis of reasoning, (i.e. start with) the part referring to the livelihood strategy in response to changes in the land and market. Now it is hard to read as a good highlight of the study.

Response 1

After careful consideration on this comment and with comments from the English editor, I have improved the title as the highlight of the paper. The new title is “Agricultural resources and trade strategies: Response to falling land-to-labor ratios in Malawi”

Question 2

In the results or conclusions section could one be tempted to make an additional paragraph discussing the results obtained from the research with those available in the literature of the subject. However, the main results were supported by literature references, and the overall reasoning of the study and conclusions are correct.

Thank you!

Response 2

I have re-organized the results and discussion sections to further compare my results with extant literature and improve the conclusion section.

I hope these changes fulfils a consideration in the journal,

Thank you.

Reviewer 3 Report

This is a well written and argued article which focuses on an important aspect of farm-level decision making in the case of smallholder farmers. While the methodological framework of the article is not in my area of expertise, I have a few suggestions for providing more detail on Malawi's rural economy. It is up to the author whether or not to incorporate these into the article.

-- Land: While Table 2 (summary statistics, p.10) possibly provides some of this data, it would be useful to get a better sense of land distribution in Malawi. What is considered a small farm in Malawi and what percentage of farm households can be classified as such (in 2016)? It would also be useful if a brief definition is provided for 'customary tenure.' This can be added to the Background (Section 1.1).

-- Gender: The article mentions female headed households in its Results [p.15] which is useful. Can a sentence of two on gender-related findings also be added to the Discussion and Conclusion sections?

How does the matriarchal system in Malawi possibly shape the renting land / casual labor decision? Maybe a mention of its possible effect (or lack of effect) can be added to the Background section, if it would help contextualize gender relations in Malawi?

A list of control variables on p.14 mentions 'share of male labor.' Was this considered as a variable?

-- It might be useful to also present the results as a chart or table. This could help quick visualization and also draw the reader into the details of the Results.

-- Should the theoretical framework and hypothesis (Section 1.2) be part of Materials and Methods, rather than Introduction?

Author Response

General comment

This is a well written and argued article which focuses on an important aspect of farm-level decision making in the case of smallholder farmers. While the methodological framework of the article is not in my area of expertise, I have a few suggestions for providing more detail on Malawi's rural economy. It is up to the author whether or not to incorporate these into the article.

General Response

Thank you for taking the time to review the paper and for giving me a chance to revise and resubmit.

After carefully considering all the comments from reviewers, I have improved the title as the highlight of the study. The new title is “Agricultural resources and trade strategies: Response to falling land-to-labor ratios in Malawi”.

Question 1

-- Land: While Table 2 (summary statistics, p.10) possibly provide some of this data, it would be useful to get a better sense of land distribution in Malawi. What is considered a small farm in Malawi and what percentage of farm households can be classified as such (in 2016)? It would also be useful if a brief definition is provided for 'customary tenure.' This can be added to the Background (Section 1.1).

Response 1

In the background section, I have added information on land distribution, holding sizes, classification of small- and large-scale farms, and land governance systems. Furthermore, I have elaborated more on customary tenure system in Malawi.   

Question 2

-- Gender: The article mentions female-headed households in its Results [p.15] which is useful. Can a sentence of two on gender-related findings also be added to the Discussion and Conclusion sections? How does the matriarchal system in Malawi possibly shape the renting land / casual labor decision? Maybe a mention of its possible effect (or lack of effect) can be added to the Background section, if it would help contextualize gender relations in Malawi?

Response 2

By defining the customary tenure system, I have included a statement on patrilineal and matrilineal systems of inheritance although this does not directly translate to patriarchal and matriarchal systems of resource control at the farm household level in Malawi. Considering that the study controlled for female-headed households, the discussion section includes a statement on the observed effect in either factor market. However, there is a need for further studies on gender dynamics and factor markets in Malawi.  

Question 3

A list of control variables on p.14 mentions 'share of male labor.' Was this considered as a variable?

Response 3

The initial construction of the model included the “share of male labour” but to improve the model, I later dropped the variable. From the statistical summary table in the paper and the table I have presented below, I observed no significant differences in the distribution of the “share of male labor” and “total household labor” across the categories. Thus, I dropped the “share of male labor” and maintained “total household labour”. Following this, I have deleted the name from the list of control variables. 

VARIABLES

Tenant

(1)

Casual labor

(2)

t-test

(1 vs. 2)

P-value

Regular farmer (Farmed in both survey rounds) (3)

Non-regular farmer (Farmed in one survey round) (4)

Non-agricultural household (No farming in all rounds) (5)

Share of male labour

0.42 (0.01)

0.42

(0.01)

0.57

0.39 (0.01)

0.45

(0.02)

0.45

(0.01)

Question 5

-- It might be useful to also present the results as a chart or table. This could help quick visualization and also draw the reader into the details of the Results.

Response 5

Indeed, a graphical presentation would give a quick assessment of the results in the paper. However, due to the number and a combination of models, I might end up with so many graphs and still need a table of results. Therefore, I have maintained the summarized Tables 3 and 4 while presenting the full model results in the Appendix section. The summarized tables present key results in line with the stated hypotheses.

Question 6

-- Should the theoretical framework and hypothesis (Section 1.2) be part of Materials and Methods, rather than Introduction?

Response 6

I have re-organized the sections to improve the presentation of the paper. Moving the “Theoretical Framework and Hypothesis” section to the “Materials and Methods” section was confusing for me. Therefore, I have numbered the section independently and changed the “Materials and Method” section to the “Data and Estimation Methods” section for a better flow and organization of the paper. 

I hope these changes fulfils a consideration in the journal,

Thank you.

Round 2

Reviewer 1 Report

Accepted in this current version.

This manuscript is a resubmission of an earlier submission. The following is a list of the peer review reports and author responses from that submission.